# CRITIQUE-OUT-LOUD REWARD MODELS

## ABSTRACT

Traditionally, reward models used for reinforcement learning from human feedback (RLHF) are trained to directly predict preference scores without leveraging the generation capabilities of the underlying large language model (LLM). This limits the capabilities of reward models as they must reason implicitly about the quality of a response, i.e., preference modeling must be performed in a single forward pass through the model. To enable reward models to reason explicitly about the quality of a response, we introduce Critique-out-Loud (CLoud) reward models. CLoud reward models operate by first generating a natural language critique of the assistant's response that is then used to predict a scalar reward for the quality of the response. We demonstrate the success of CLoud reward models for both Llama-3-8B and 70B base models: compared to classic reward models CLoud reward models improve pairwise preference classification accuracy on RewardBench by 4.65 and 5.84 percentage points for the 8B and 70B base models respectively. Furthermore, CLoud reward models lead to a Pareto improvement for win rate on ArenaHard when used as the scoring model for Best-of-N. Finally, we explore how to exploit the dynamic inference compute capabilities of CLoud reward models by performing self-consistency decoding for reward prediction.

## 1 INTRODUCTION

In reinforcement learning from human feedback (RLHF) (Christiano et al., 2017; Nguyen et al., 2017), a reward model is trained as a proxy for human preferences. Such reward models are then used to produce a human-preference aligned generation policy. Methods to do this include RL training or generating multiple responses and selecting the highest scoring generation under the reward model. In this work, we focus on improving the performance of reward models by training them to critique responses before predicting a reward.

Generally, reward models are trained as simple LLM based classifiers of the user's prompt and the assistant's response (Stiennon et al., 2020; Ouyang et al., 2022). Importantly, the language modeling (LM) head of the underlying LLM is not used during reward modeling. We hypothesize that this limits the performance of classic reward models as they cannot explicitly reason about the quality of the response in a Chain-of-Thought (CoT) (Wei et al., 2022) like manner. Namely, without generating reasoning traces, all reasoning in classic reward models must be performed implicitly in the model within a single forward pass.

The utility of reasoning traces for preference modeling is demonstrated by the LLM-as-a-Judge framework (Zheng et al., 2023), where a scoring rubric is provided to an LLM, and the LLM reasons about how the provided response adheres to the rubric before scoring the quality of the response. While LLM-as-a-Judge provides both the ability to define preferences at inference time through the judging rubric and interpretable evaluation by inspecting the produced CoT reasoning, LLM-as-a-Judge generally under-performs classic reward models at pairwise preference classification[1].

In this work, we investigate how to leverage the language generation capabilities of LLMs to improve reward model performance. Adding the capacity for language generation to reward models enables them to explicitly reason about the quality of the input via variable inference compute in a CoT like manner. To this end, we propose Critique-out-Loud (CLoud) reward models: conditioned on the user's prompt and the assistant's response, CLoud reward models first generate a detailed critique

---

[1] https://huggingface.co/spaces/allenai/reward-bench

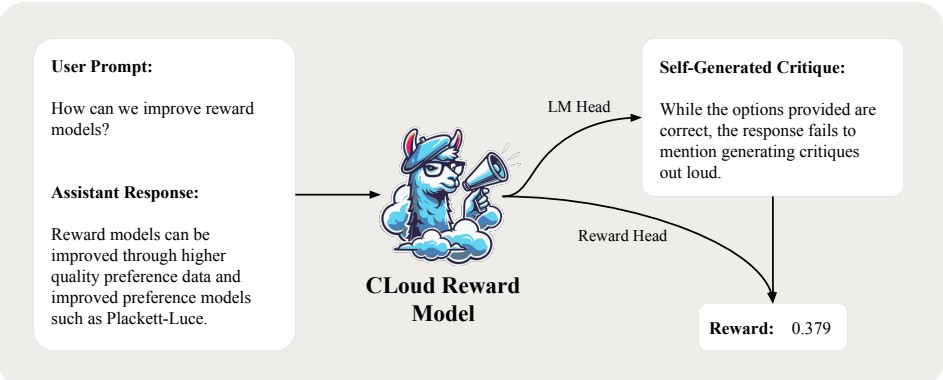

Figure 1: **Overview of CLoud reward models**. CLoud reward models augment classic reward models with a language modeling (LM) head to provide critiques in addition to a scalar reward. Given a user's prompt and an assistant's response as inputs, a CLoud reward model first generates a critique describing the quality of the assistant's response. Then, conditioned on the prompt, response, and self-generated critique, the CLoud reward head produces a scalar reward.

about how well the response answers the user's query. Then, as a function of the user's prompt, the assistant's response, and the self-generated critique, the CLoud reward model produces a scalar reward for the quality of the response. We present an overview of CLoud reward models in Figure 1. By introducing language generation to classically trained reward models, our work provides the groundwork to unify classic reward models and LLM-as-a-Judge and inherits the advantages of both methods. To train CLoud reward models we assume access to a preference dataset composed of prompts, responses, and oracle critiques of the responses. We train CLoud reward models to both generate critiques by supervised finetuning (SFT) on the oracle critiques and to produce scalar rewards based on the Bradley-Terry (BT) preference model (Bradley & Terry, 1952).

We also explore how to exploit the stochasticity in critique generation via multi-sample inference techniques to improve reward modeling performance. Specifically, we investigate self-consistency (Wang et al., 2023a) for CLoud reward models and sample multiple (critique, reward) predictions before marginalizing across critiques to produce a better estimate of the reward.

**Contributions**   Our work makes the following contributions:

- We introduce Critique-out-Loud (CLoud) reward models: reward models that are trained to explicitly reason about the quality of responses before scoring them. Through adding critique capabilities to reward models, CLoud lays the groundwork for unifying reward models and LLM-as-a-Judge.

- We demonstrate that CLoud reward models improve pairwise preference classification accuracy on RewardBench by up to 4.65 and 5.84 percentage points for the 8B and 70B base models respectively (Figure 3). Additionally, we show that CLoud reward models lead to a Pareto improvement for win rate on ArenaHard when used as the scoring model for Best-of-N (Figure 4).

- We ablate an important design choice in the training of CLoud reward models: on versus off-policy training. We show that on-policy training is essential for the success of CLoud reward models for both preference classification and for BoN (Figures 6 and 7).

- We investigate self-consistency over critiques as a method to trade added inference compute for better reward modeling. We demonstrate that self-consistency over the critiques improves pairwise preference classification accuracy for reasoning tasks by up to 0.70 and 0.49 percentage points for the 8B and 70B models respectively (Figure 7).

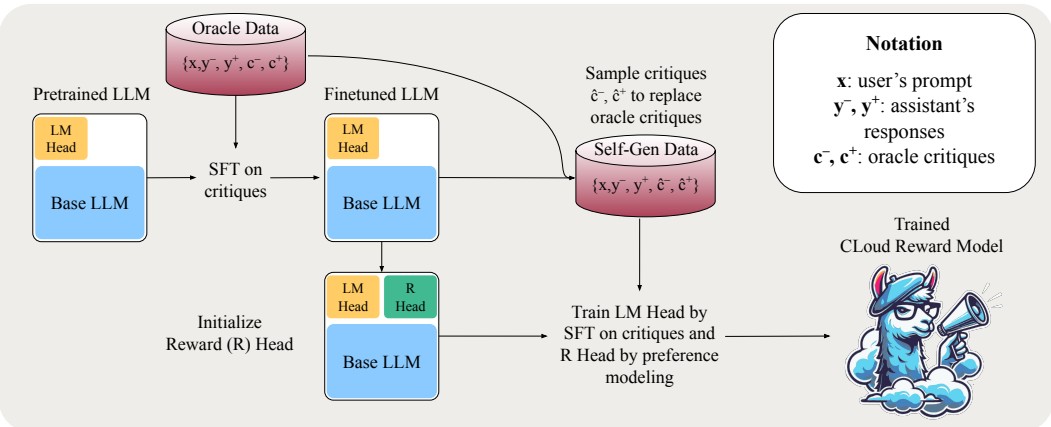

Figure 2: **Overview of training CLoud reward models.** CLoud reward models are trained using a dataset that consists of user prompts, chosen and rejected assistant responses, and critiques of the response quality generated by an oracle. We first train a pretrained LLM to generate critiques for the responses by finetuning on the oracle critiques. We then rebuild the dataset by replacing oracle critiques with critiques generated by the finetuned model. Finally, we initialize a scalar reward head on top of the finetuned model and train on the new dataset composed of self-generated critiques to minimize both a language modeling and a preference modeling loss.

## 2 METHODS

In this section, we review how classic reward models that model human preferences are trained and we then extend this methodology to training CLoud reward models. We also detail how CLoud reward models are used to score samples at inference time using both standard and self-consistency decoding. Note, we will refer to the trunk of a pretrained LLM before the final language modeling layer as the *base model* and the linear or shallow multi-layer perceptrons (MLPs) that operate on the output of the base model as *heads*.

### 2.1 CLASSIC REWARD MODELS

Typically, classic reward model consists of a base model and a shallow MLP reward head. Its parameters are $(\theta_B, \theta_R)$, where $\theta_B$ and $\theta_R$ are the parameters of the base model and reward head respectively. Given a user prompt $x$ and an assistant response $y$, the classic reward model predicts a scalar reward score $\hat{R} = r_{\theta_B,\theta_R}(x, y)$. A classic reward model is initialized from a pretrained base model and a randomly initialized reward head and then trained on a dataset of $N$ examples, $D = \{(x, y^-, y^+)_i\}_{i=1}^N$. Here, $x$ is a user's prompt and the $y$s are two different assistant responses to the prompt: $y^+$ is the chosen or preferred response and $y^-$ is the rejected response as judged by a human or a more powerful model. Reward models are trained to predict a higher reward for $y^+$ than for $y^-$ under the Bradley-Terry model (Bradley & Terry, 1952). This is achieved by minimizing:

$$\mathcal{L}_{RM}(\theta_B, \theta_R, D) = -\mathbb{E}_{(x,y^-,y^+)\sim D}[\log(\sigma(r_{\theta_B,\theta_R}(x, y^+) - r_{\theta_B,\theta_R}(x, y^-)))]$$

where $\sigma(*)$ is the sigmoid function.

### 2.2 CLOUD REWARD MODELS

In addition to the base model and reward head, CLoud reward models preserve the language modeling head of the original pretrained LLM and are defined by parameters $\theta = (\theta_B, \theta_{LM}, \theta_R)$ where $\theta_{LM}$ are the parameters of the language modeling head. CLoud reward models extend classic reward models by first generating a critique of the assistant's response and then predicting a scalar reward conditioned on the critique (depicted in Figure 1). Formally, given a user prompt $x$ and assistant response $y$ we first sample a critique $\hat{c} \sim p(*|x, y; \theta_B; \theta_{LM})$ and then predict a reward conditioned on the prompt, the response, *and the critique*: $\hat{R} = r_{\theta_B;\theta_R}(x, y, \hat{c})$.

**Training CLoud reward models.** CLoud reward models are trained with a dataset of $N$ examples, $D = \{(x, y^-, y^+, c^-, c^+)_i\}_{i=1}^N$, where we introduce oracle critiques, $c^-, c^+$, of the rejected and chosen responses $y^-, y^+$ respectively. The critiques are reasoning traces that provide feedback on the weaknesses of the responses and strategies for improving them. While ideally $c^-, c^+$ would be human critiques of the responses, we use critiques generated by a more powerful model, specifically Llama-3.1-405B-Instruct (Dubey et al., 2024), to approximate human critiques as done in prior work (Bai et al., 2022b; Dubois et al., 2024). Further details on how these oracle critiques are generated are provided in Appendix A.

To train CLoud reward models we: (1) train the base model and LM head to generate critiques via supervised finetuning on the oracle critiques, (2) replace oracle critiques in the dataset with critiques generated by the finetuned model, and (3) train a reward head conditioned on self-generated critiques. We choose to train the reward head on self-generated critiques as to minimizes the distribution shift in the critiques seen by the reward head between training and inference when oracle critiques are not available. We present an overview of CLoud reward model training in Figure 2.

Before formally detailing the steps of CLoud reward model training, we introduce the following objectives. First, we modify $\mathcal{L}_{RM}$ to work with CLoud reward models as:

$$\mathcal{L}_{RM}(\theta_B, \theta_R, D) = -\mathbb{E}_{(x,y^-,y^+,c^-,c^+) \sim D}[\log(\sigma(r_{\theta_B, \theta_R}(x, y^+, c^+) - r_{\theta_B, \theta_R}(x, y^-, c^-)))]$$

where the reward estimator $r_\theta(x, y, c)$ is now also conditioned on a critique $c$. Next, we introduce the critique SFT loss, which is the negative log likelihood of the rejected and chosen critiques:

$$\mathcal{L}_{SFT}(\theta_B, \theta_{LM}, D) = -\mathbb{E}_{(x,y^-,y^+,c^-,c^+) \sim D}\big[ \sum_{c_t^- \in c^-} \log p(c_t^- | x, y^-, c_{<t}^-; \theta_B, \theta_{LM})$$
$$+ \sum_{c_t^+ \in c^+} \log p(c_t^+ | x, y^+, c_{<t}^+; \theta_B, \theta_{LM})\big]$$

where $c_{<t} = c_1, ..., c_{t-1}$ is the prefix of the critique up to the $t^{th}$ token. Finally, we introduce a joint SFT and RM loss as:

$$\mathcal{L}_{CLoud}(\theta_B, \theta_{LM}, \theta_R, D) = \mathcal{L}_{RM}(\theta_B, \theta_R, D) + \lambda \cdot \mathcal{L}_{SFT}(\theta_B, \theta_{LM}, D)$$

where $\lambda$ is a hyperparameter that weights the contribution of the language modeling loss.

To train CLoud reward models we first train $(\theta_B, \theta_{LM})$ to generate critiques by minimizing $\mathcal{L}_{SFT}$ on the oracle critiques in the oracle dataset $D$ and obtain parameters $\tilde{\theta}_B, \tilde{\theta}_{LM}$. We use the finetuned model to modify our dataset with self-generated critiques that replace the oracle critiques. Specifically, given the $i^{th}$ element from our dataset $(x, y^-, y^+, c^-, c^+) = D_i$ we sample new critiques $\hat{c}^- \sim p(*|x, y^-; \tilde{\theta}_B, \tilde{\theta}_{LM})$ and $\hat{c}^+ \sim p(*|x, y^+; \tilde{\theta}_B, \tilde{\theta}_{LM})$. We then construct a new dataset as $D_{self,i} = (x, y^-, y^+, \hat{c}^-, \hat{c}^+)$. Finally, we obtain our CLoud reward model parameters by training on our self-critique dataset:

$$\theta_B^*, \theta_{LM}^*, \theta_R^* = \argmin_{\theta_B, \theta_{LM}, \theta_R} \mathcal{L}_{CLoud}(\theta_B, \theta_{LM}, \theta_R, D_{self})$$

where $\theta_B, \theta_{LM}$ are initialized to the finetuned parameters $\tilde{\theta}_B, \tilde{\theta}_{LM}$. We train CLoud reward models on the joint loss $\mathcal{L}_{CLoud}$ instead of $\mathcal{L}_{RM}$ to preserve the critique generation capability of the CLoud reward model and to prevent over-fitting to solely producing reward scores.

**Self-consistent reward scores.** Self-consistency (Wang et al., 2023a) is an inference technique for computing the maximum marginal likelihood answer by sampling multiple (`reasoning`, `answer`) tuples and marginalizing over the reasoning traces. It provides a simple method to improve performance at the cost of added inference compute. In this work, we leverage self-consistency to provide a better estimate of the reward by marginalizing over critiques.

Given a prompt $x$ and response $y$ to score, we first sample $N$ critiques $c_1, c_2, \ldots, c_N \sim p(*|x, y; \theta_B; \theta_{LM}; \tau)$, where $\tau$ is the sampling temperature. For each critique, we predict a reward as $\hat{R}_i = r_{\theta_B, \theta_R}(x, y, c_i)$. We then estimate the true reward as the mean of the individual rewards.

## 3 RESULTS

In this section we detail our experimental setup and provide evaluations of CLoud reward models in the pursuit of answering the following research questions:

**RQ1:** How does CLoud impact performance, both in terms of preference classification and the quality of the policy achievable by maximizing its reward (Figures 3 and 4)?

**RQ2:** Is it necessary to train CLoud reward models in an on-policy manner via training on self-generated critiques (Figures 5 and 6)?

**RQ3:** Does self-consistency decoding benefit the performance of CLoud reward models, and if so, under what distribution of inputs (Figures 7 and 8)?

### 3.1 SETUP

**Training.** As base models we experiment with the Llama-3 family of models (Dubey et al., 2024). Specifically, we train reward models starting from the Llama-3-8B and Llama-3-70B base models. For both classic and CLoud reward models we perform a hyperparamater sweep over learning rate and number of training epochs. For CLoud reward models we additionally sweep the SFT loss weight $\lambda$. We provide further details on our hyperparamater sweep in Appendix B. We use a cosine learning rate schedule (Loshchilov & Hutter, 2017) with a warmup duration of 5% and a final decay factor of 1%. Additionally, we use the Adam optimizer with decoupled weight decay (Loshchilov & Hutter, 2019) with parameters $\beta_1 = 0.9, \beta_2 = 0.95, \epsilon = \text{1e-10}$. We train each model with two random seeds. All models are trained using `Nvidia H100` gpus and training is conducted using MosaicML Composer (MosaicML, 2021).

**Data.** For training data we use a mix of the prompts from the UltraFeedback (Cui et al., 2023) and UltraInteract (Yuan et al., 2024a) datasets. Together, UltraFeedback and UltraInteract contain a diverse collection of prompts covering topics such as general chat, instruction following, reasoning, etc. As UltraInteract is almost twice the size of UltraFeedback, we first uniformly sub-sample UltraInteract to be the same size as UltraFeedback before merging the prompts in the two datasets.

We replace the chosen and rejected responses in the merged Ultra dataset with responses from Llama-3-8B-Instruct. Specifically, for each prompt we sample two responses from Llama-3-8B-Instruct and assign chosen and rejected labels through a pairwise judgement using Llama-3.1-405B-Instruct as an oracle preference model. To perform the pairwise judgement we use the pairwise judgement prompt from ArenaHard (Li et al., 2024b). We refer to the dataset composed of prompts from UltraFeedback and UltraInteract but with responses from Llama-3-8B-Instruct as UltraLlama.

After labeling the chosen and rejected responses, we generate oracle critiques for each of the chosen and rejected responses using Llama-3.1-405B-Instruct as the oracle. We do so by prompting the oracle model to provide detailed, step-by-step feedback about the correct and incorrect elements of each response. The prompt we use to generate oracle critiques can be found in Appendix A.

**Evaluation.** We evaluate the quality of reward models on both pairwise preference classification accuracy and Best-of-N (BoN) win rate.

We evaluate pairwise preference classification on the RewardBench (Lambert et al., 2024) evaluation suite, which is composed of $2,985$ examples and is organized into Chat, Chat-Hard, Safety, and Reasoning categories. Each example contains a prompt and a chosen and rejected response and a reward model is evaluated as to whether it predicts a greater reward for the chosen response. In addition to the accuracy on each category, we report the average accuracy across all categories.

We evaluate BoN win rate performance on ArenaHard (Li et al., 2024b), an open-ended generation benchmark consisting of five hundred prompts meant to reflect high-quality, real-world use cases of LLMs. To preform BoN, for each user query we sample $N$ potential responses from Llama-3-8B-Instruct. Then, for a given reward model, we compute its reward on each of the $N$ responses and select the "best" response as the response with the highest reward. To evaluate the performance of the BoN generations, we use the ArenaHard eval harness to compute the win rate of the BoN generations as compared to greedy generations from Llama-3.1-70B-Instruct using Llama-3.1-405B-Instruct as the judge. We evaluate BoN at $N = 2, ..., 16$ where responses are generated at a temperature of $1.0$.

**CLoud vs. Classic on RewardBench**

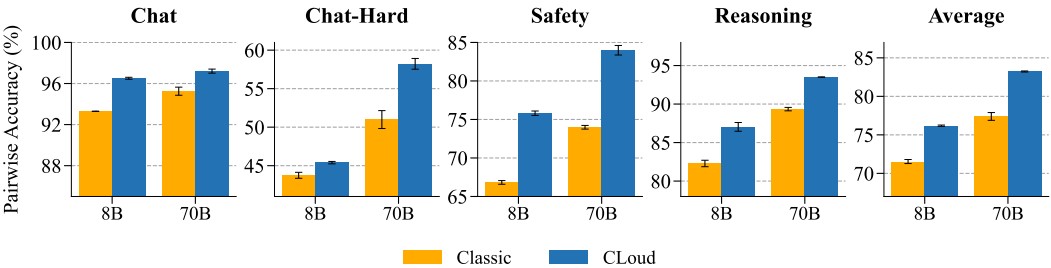

Figure 3: **Comparing pairwise preference classification accuracy of classic and CLoud reward models on RewardBench.** Pairwise preference classification accuracy measures if the reward model correctly classifies the chosen and rejected responses. At both the 8B and 70B model scales, CLoud reward models significantly outperform classic reward models on all categories. This leads to a large increase in average accuracy for CLoud reward models.

**CLoud vs. Classic on ArenaHard**

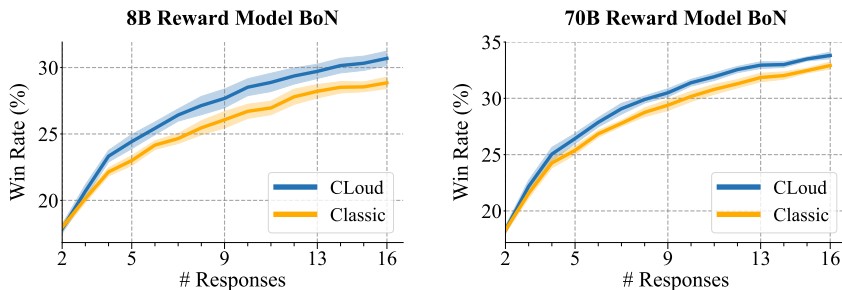

Figure 4: **Comparing Best-of-N (BoN) win rates using classic and CLoud reward models on ArenaHard.** The shaded area represents $\pm 1$ standard error from the mean. To perform BoN we sample a given number of responses, compute the reward for each response, and then select the response with the highest score from the reward model. BoN serves as a proxy for the quality of a policy achievable under a reward model. At both model sizes, CLoud is a Pareto improvement meaning that it produces an equal or significantly higher win rate BoN policy than the classic reward model. For Best-of-16, CLoud improves the win rate by 1.84 and 0.89 percentage points for the 8B and 70B models respectively.

Additionally, we found high variance in win rate based on the set of responses sampled, and as such, we average the BoN win rate for each $N$ over four different seeds of responses.

## 3.2    COMPARING CLASSIC AND CLOUD REWARD MODELS

**RQ1: Do critiques improve performance?**    To test whether critiques improve reward model performance we first compare classic reward models to CLoud reward models on RewardBench (Figure 3). On RewardBench, we find that across both model sizes, CLoud reward models lead to large gains in pairwise preference accuracy, significantly outperforming the corresponding classic reward model on all categories. On average, we find that CLoud reward models outperform classic reward models by 4.65 and 5.84 percentage points for the 8B and 70B base models respectively. Furthermore, on both the chat and safety categories, the 8B CLoud reward model even outperforms the 70B classic reward model. To better understand the critiques generated by CLoud reward models, we present critiques from examples in RewardBench in Appendix D.

We next evaluate BoN with classic and CLoud reward models in Figure 4. We find that for all model sizes BoN with CLoud reward models is a Pareto improvement over BoN with classic reward

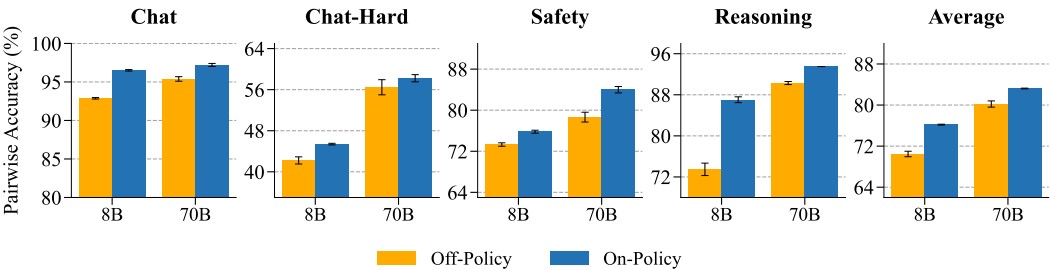

Figure 5: **Comparing pairwise classification accuracy between CLoud reward models trained on-policy and off-policy on RewardBench.** In this experiment, we ablate the importance of self-generating critiques for training CLoud reward models. For CLoud off-policy, the reward head is trained on the oracle critiques. For CLoud on-policy, the reward head is trained on self generated critiques. For both model sizes, training in an off-policy manner leads to a significant drop in performance. This highlights the importance of matching the distribution of critiques seen by the reward head during training and inference.

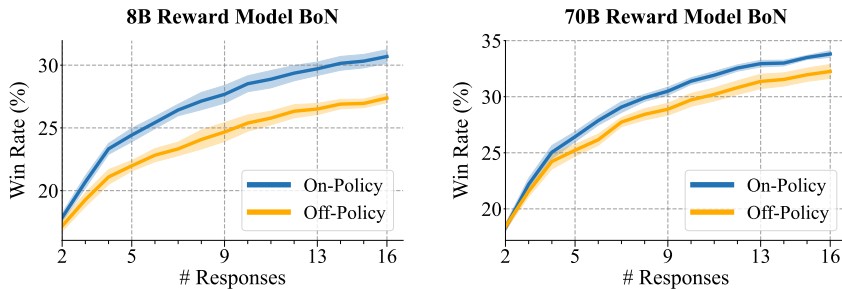

Figure 6: **Comparing BoN win rates on ArenaHard of CLoud reward models trained on and off-policy.** The shaded area represents ±1 standard error from the mean. At both model sizes, we find that the win rate from performing BoN with CLoud reward models trained on-policy is a Pareto improvement over those trained off-policy. For Best-of-16, training off-policy decreases the win rate by 3.31 and 1.56 percentage pooints for the 8B and 70B models respectively, highlighting the importance of training on-policy CLoud reward models for defining better downstream policies.

models, meaning that for each number of responses, the BoN win rate with CLoud is equal to or better than that of classic. Selecting from sixteen responses with CLoud reward models leads to a win rate improvement of 1.84 and 0.89 percentage points as compared to classic reward models for the 8B and 70B base models respectively.

These results suggest that adding the capability for the reward models to generate critiques leads to significant performance gains in preference modeling. Furthermore, the improvements in preference modeling transfer to improving the quality a generation policy.

**RQ2: Is on-policy training necessary?** In our setup for training CLoud reward models, we modify the original dataset by replacing all oracle critiques with self-generated critiques before training the reward head. We do so to mitigate the distribution shift between the critiques the reward head is trained on and the critiques available at inference. To determine whether this on-policy training is necessary, we train an off-policy CLoud reward model by training on oracle critiques instead of self-generated critiques. Namely, we train our off-policy CLoud reward model by minimizing $\mathcal{L}_{CLoud}$ over the original oracle critique labeled dataset, instead of the self-critique dataset.

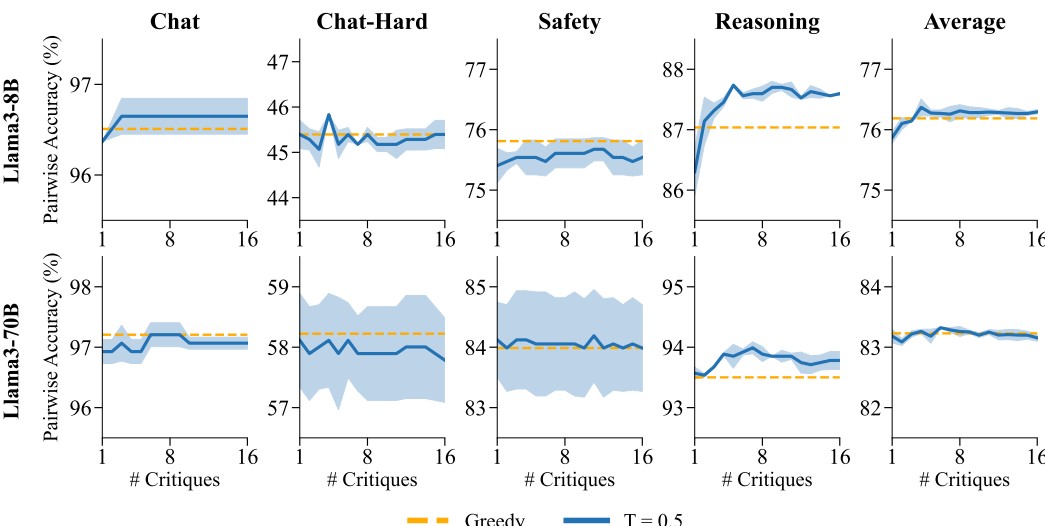

Figure 7: **Effect of self-consistency decoding on performance of CLoud reward models on RewardBench.** The shaded area represents $\pm 1$ standard error from the mean. For each prompt and response input, we sample multiple (critique, reward) tuples from the CLoud reward model and average the reward over the critiques to provide a better estimate of the reward. While self-consistency does not improve the performance for most categories, on the reasoning category it is an effective method to trade added inference compute for increased performance.

We plot the pairwise preference modeling accuracy for on-policy and off-policy CLoud reward models in Figure 5. CLoud trained on-policy significantly outperforms CLoud trained off-policy on all categories except for chat-hard at the 70B base model scale. Off-policy training leads to a 5.60 and 3.03 percentage point drop in average performance for the 8B and 70B base models respectively.

We plot the BoN win rate for on-policy and off-policy CLoud reward models in Figure 6. We find that CLoud reward models trained on-policy are a Pareto improvement in BoN win rate over models trained off-policy. Specifically, selecting from sixteen responses with the off-policy reward model leads to a 3.31 and 1.56 percentage point decrease in win rate as compared to the on-policy reward models for the 8B and 70B base models respectively.

These results suggests that training CLoud reward models in an on-policy manner is necessary to achieve strong performance.

## 3.3 SELF-CONSISTENCY FOR CLOUD REWARD MODELS

**RQ3: Do CLoud reward models benefit from added inference compute?** To test whether CLoud reward models benefit from added inference compute, we examine how accuracy changes when using self-consistency decoding. For each response in RewardBench, we sample up to sixteen critiques at a temperature of 0.5. We plot the performance of self-consistency decoding on Reward-Bench for 8B and 70B CLoud reward models in Figure 7. We find that reasoning is the only category that benefits from additional inference compute in the form of self-consistency. Specifically, we find that self-consistency leads to an improvement in preference classification accuracy of up to 0.70 and 0.49 percentage points for the 8B and 70B base models respectively. Furthermore, that we do not see a gain in self-consistency for non-reasoning categories agrees with the results of Lee et al. (2024). We also evaluate the effect of self-consistency reward modeling for BoN on ArenaHard. Unlike RewardBench, we do not find any gain in BoN win rate from self-consistent rewards. For the sake of brevity we present these results in Appendix C. Our self-consistency results provide initial evidence that in specific situations, added inference compute can improve the performance of CLoud

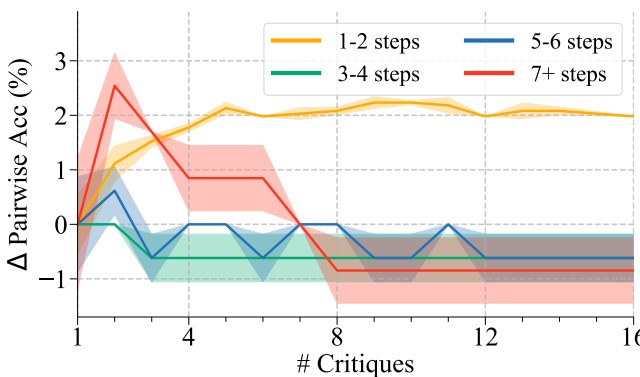

Figure 8: **The dependence of performance gain from self-consistency on the number of reasoning steps in the assistant's response.** We find that prompts that require only 1-2 reasoning steps observe a significant lift in performance as the number of critiques increases, while all other groups of reasoning steps actually degrade in performance past eight critiques. This result shows that self-consistency with CLoud reward models enables better judgement of response quality for short horizon tasks.

reward models, but that it is important to know the distribution of tasks being scored as not all task categories benefit.

**RQ3: When is self-consistency useful?** To better understand when self-consistency improves pairwise preference classification accuracy, we investigate the effect that a response's reasoning horizon has on self-consistency's performance. We do so on the reasoning split of RewardBench and we approximate the number of reasoning steps required for a problem as the average number of sentences in the chosen and rejected response. We bin the number of reasoning steps as 1-2, 3-4, 5-6, and 7+ steps. We plot the performance gain from self-consistency grouped by reasoning steps for CLoud 8B on the reasoning split of RewardBench in Figure 8. We find that only problems requiring 1-2 steps see a consistent gain in pairwise preference classification accuracy as the number of critiques increases, while the performance on problems with a greater number of steps actually decreases after eight critiques. This result provides initial evidence that CLoud + self-consistency may be a strong combination when evaluating solutions with short reasoning horizons.

## 4 RELATED WORK

**Classic reward models.** In RLHF, reward models have traditionally modeled human preferences after ranking models such as the Bradley-Terry (BT) model (Bradley & Terry, 1952; Ouyang et al., 2022; Bai et al., 2022b;a; Dubey et al., 2024) or the Plackett-Luce model (Plackett, 1975; Luce, 1959; Zhu et al., 2023). Recent work has showed shortcomings of these models when handling intransitive preferences (Munos et al., 2023; Swamy et al., 2024). Another line of work directly models the probability of one response being preferred over the other (Jiang et al., 2023; Zhao et al., 2023; Liu et al., 2023; Dong et al., 2024; Swamy et al., 2024). Finally, another line of work aims to model rewards over multiple objectives (Wang et al., 2023b; 2024b;a). The improvements of CLoud reward models are orthogonal to the above methods as CLoud is agnostic to the preference modeling objective. Future work should explore the composition of CLoud reward models with more complex reward model objectives. Recently, Yang et al. (2024) proposed to maintain the LM head of a reward model, and train the LM head on the chosen and rejected responses as a form of regularization. While CLoud also maintains and further trains the LM head, we do so for the purpose of generating critiques.

**Critique-based feedback.** There is a large body of work that concerns providing feedback in the form of natural language critiques. In settings where oracle critiques or signals for critique quality do not exist, past works have explored using the model itself to generate critiques that are then either

referenced to improve generation quality or directly leveraged as a preference signal (Bai et al., 2022b; Shinn et al., 2024; Ganguli et al., 2023; Madaan et al., 2024; Yuan et al., 2024b; Ramji et al., 2024; Kim et al., 2024). While our work also leverages self-generated critiques at inference, our work differs in that we aim to train better reward models when human preference data is available as opposed to bootstrapping preferences from the model itself. Lee et al. (2024) extend self-generated critiques for preference modeling by leveraging additional inference compute via self-consistency to improve preference modeling performance. While they find that self-consistency does not help for the tasks they examine, we demonstrate that self-consistency does help for reward modeling on reasoning tasks (Section 3.3).

Previous works have also explored the setting where oracle critiques are available for training. Saunders et al. (2022); Akyurek et al. (2023) teach an LLM to critique by performing SFT on oracle critiques and McAleese et al. (2024) teach an LLM to critique by performing RLHF on human-labeled critique preferences. Other works leverage access to oracle feedback (e.g., human, error traces, etc.) to generate refined answers conditioned on the critiques (Gao et al., 2023; Scheurer et al., 2023; Chen et al., 2024; Gou et al., 2024). Our work differs from the above as our goal is to leverage critiques to train better reward models. Most similar to our work is that of Ye et al. (2024) which explores improving reward model performance by training reward models on human preferences conditioned on critiques generated by other models. While their work similarly demonstrates the advantages of reward scores conditioned on critiques, our work differs in that we investigate training the reward model to generate its own critiques. As such, CLoud reward models can perform inference without requiring access to a larger model.

**LLM-as-a-Judge.** In the LLM-as-a-Judge framework, an LLM scores responses based on a user provided grading rubric (Gilardi et al., 2023; Huang et al., 2023; Zheng et al., 2023; Kim et al., 2023; Li et al., 2024a). While similar to methods above such as Constitutional AI, LLM-as-a-Judge differs in that the objective is to evaluate responses, not revise responses. Similar to CLoud reward models, LLM-as-a-Judge produces chain-of-thought reasoning about how the grading rubric applies to the response before producing a score. However, CLoud differs from LLM-as-a-Judge as our models maintain a scalar reward head and as such can be trained according to classic reward modeling objectives such as the BT model. Our work takes a first step towards unifying the classic reward model and LLM-as-a-Judge methods for preference modeling. Future work should investigate how the human crafted grading rubrics used in LLM-as-a-Judge can be integrated with the critique process of CLoud reward models.

## 5 Conclusion

In this work, we propose Critique-out-Loud (CLoud) reward models which preserve the language modeling capabilities of the underlying LLM while additionally training a scalar reward head. To perform inference with CLoud reward models, we first sample a critique of the response from the reward model before predicting a scalar reward. Through generating critiques, CLoud reward models can explicitly reason about the quality of a response while classic reward models must reason implicitly. We demonstrate on the RewardBench evaluation suite that, as compared to classic reward models, CLoud reward models can improve average pairwise preference modeling accuracy by up to 4.64 and 5.84 percentage points for 8B and 70B base models respectively. Similarly, we demonstrate that performing Best-of-N decoding with CLoud reward models is a Pareto improvement over classic reward models for ArenaHard win-rate. We further investigate how CLoud reward models can leverage additional inference compute via multi-sample decoding strategies. Specifically, we evaluate self-consistency decoding for CLoud reward models where we marginalize over sampled critiques to provide a better estimate of the reward. We find that CLoud reward models only benefit from self-consistency on reasoning problems and demonstrate that self-consistency is predominantly useful when assigning rewards to responses with short reasoning horizons. CLoud reward models establish a new paradigm for reward models by unifying language generation with preference modeling and open new avenues for improving reward models through variable inference compute.

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

> **Oracle Critique Generation Prompt**
>
> **System Prompt:**
>
> Please act as an expert in providing feedback. You will be given a user's prompt and an assistant's response to the prompt. Your job is to think step by step and provide a thoughtful and detailed analysis of how well the response answers the user's query.
>
> When providing feedback, consider if the assistant's answer is helpful, relevant, and concise. Helpful means the answer correctly responds to the prompt or follows the instructions. Relevant means all parts of the response closely connect or are appropriate to what is being asked. Concise means the response is clear and not verbose or excessive. Finally, identify any missing important information in the assistants' answers that would be beneficial to include when responding to the user prompt.
>
> It is not necessary to be polite when providing feedback. Think deeply to identify all the good and bad parts of the answer.
>
> Do not numerically score the response when evaluating it. Think deeply and provide linguistic feedback and analysis only.
>
> **User Prompt**
>
> Here is the user's prompt and the assistant's response.
>
> <|User Prompt |>{user's prompt}<|End of User Prompt|>
>
> <|The Start of the Assistant's Answer|>{answer}<|The End of the Assistant's Answer|>

Figure 9: Prompt used to construct oracle critiques during dataset construction.

## A  GENERATING ORACLE CRITIQUES

To approximate human critiques when constructing our dataset we generate oracle critiques using Llama-3.1-405B-Instruct (Dubey et al., 2024). The exact prompt we use to generate oracle critiques is displayed in Figure 9.

## B  TRAINING HYPERPARAMETER SWEEP

For fair comparison, we sweep over the parameters of both the classic and CLoud reward models. For 8B models, we first evaluate learning rates of `1e-6, 5e-6,` and `1e-5`. Then, using the best learning rate, evaluate training for 1, 2, and 3 epochs. For CLoud reward models we also evaluate the SFT loss weight $\lambda$ at $\frac{3}{4}$, 1, and $\frac{5}{4}$. For the 8B base model, we find the best performing parameters for classic reward models are $\{$`lr=5e-6, epochs=2`$\}$ and that the best performing parameters for CLoud reward models are $\{$`lr=1e-6, epochs=1`$, \lambda = \frac{5}{4}\}$. We perform a similar sweep for 70B models, evaluating learning rates of `1e-6,` and `5e-6`. All 70B models are trained for only 1 epoch. For CLoud reward models we again evaluate $\lambda$ at $\frac{3}{4}$, 1, and $\frac{5}{4}$. We find the best performing parameters for classic reward models are $\{$`lr=5e-6`$\}$ and the best performing parameters for CLoud reward models are $\{$`lr=1e-6,` $\lambda = \frac{3}{4}\}$.

## C  SELF-CONSISTENCY FOR BON

In this section we explore the effect of self-consistency decoding for CLoud reward models on BoN win rate for ArenaHard. For each number of responses that we are performing BoN over, we sample sixteen critiques at a temperature of 0.5 that we average the reward over. We plot the BoN win rate

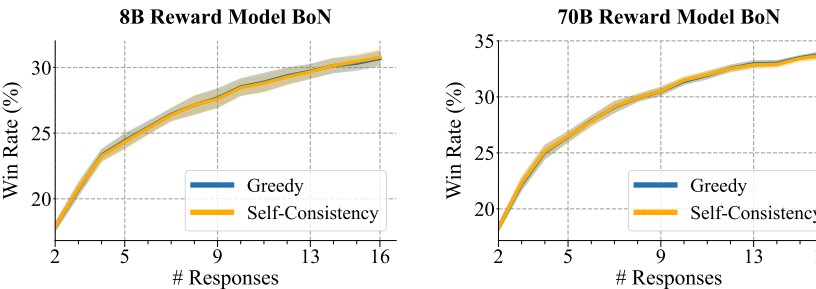

Figure 10: **Comparing BoN win rates on ArenaHard of CLoud reward models using greedy and self-consistency decoding.** The shaded area represents ±1 standard error from the mean. To perform self-consistency for BoN, we sample sixteen critiques of each response and average the reward across critiques. At both model sizes, we find no difference in BoN win rates between greedy and self-consistency decoding.

for greedy and self-consistency decoding in Figure 10. For both model sizes, we find that the BoN win rate is the same for greedy reward scoring and for self-consistency reward scoring, meaning that there there is no observed advantage in the BoN policy for performing self-consistency to predict the reward on ArenaHard.

## D    EXAMPLE REWARD PREDICTIONS ON REWARDBENCH

In this section we present examples of the reward prediction process for CLoud reward models on RewardBench. We randomly sample an example from the chat and reasoning categories, and evaluate both the 8B and 70B CLoud reward models on these examples. For each example, we present the user's query, the preferred and non-preferred responses, the corresponding critiques, and the predicted rewards. We present the 8B CLoud critiques on the preferred and non-preferred chat responses in Figures 11 and 12 respectively, and the 8B CLoud critiques on the preferred and non-preffered reasoning responses in Figures 13 and 14 respectively. We present the 70B CLoud critiques on the preferred and non-preferred chat responses in Figures 11 and 12 respectively, and the 70B CLoud critiques on the preferred and non-preffered reasoning responses in Figures 13 and 14 respectively.

**User query**

I've recently started playing the turn-based strategy game Into the Breach. I have read online that the game is considered to have 'perfect information'. What is meant by 'perfect information' in this context?

**Preferred Response**

In the context of Into the Breach and other strategy games, "perfect information" refers to a situation where all players have complete knowledge of the game state, including information about the enemy's units, abilities, and intentions. In other words, there is no hidden information that could affect the player's decision-making.

In Into the Breach, this means that you, as the player, have access to all the information you need to make informed decisions about how to deploy your units, which enemies to target, and how to plan your attacks. This includes knowledge of the enemy's unit types, their strengths and weaknesses, their movement patterns, and their attack ranges.

This is in contrast to games with "imperfect information," where players may not have access to all the information they need to make optimal decisions. For example, in a game with fog of war, players may not be able to see enemy units that are hiding behind walls or in other areas of the map. In games with imperfect information, players must use their intuition, deduction, and experience to make informed decisions despite the lack of complete information.

The designers of Into the Breach have chosen to design the game with perfect information to create a focus on strategic planning and decision-making, rather than on reacting to unexpected events or hidden information. This allows players to concentrate on optimizing their moves and planning for the long-term, rather than worrying about sudden surprises or unknown threats.

**Critique of Preferred Response**

The assistant's response provides a clear and concise explanation of what "perfect information" means in the context of Into the Breach. The answer accurately defines perfect information as a situation where all players have complete knowledge of the game state, including information about the enemy's units, abilities, and intentions.

The response effectively uses a contrast with imperfect information to illustrate the concept, highlighting the differences between games with and without perfect information. The example of fog of war is a good choice, as it is a common concept in strategy games that is easily relatable to the concept of perfect information.

The explanation of how perfect information affects the gameplay in Into the Breach is also clear and relevant. The assistant explains that the game's design choice of perfect information allows players to focus on strategic planning and decision-making, rather than reacting to unexpected events or hidden information.

However, the response could benefit from a more detailed explanation of how perfect information is achieved in Into the Breach. For example, the assistant could explain how the game's interface and mechanics provide players with all the necessary information to make informed decisions. This could include details about the game's user interface, such as the use of overhead maps, unit health bars, and other visual indicators that provide players with a clear understanding of the game state.

Additionally, the response could benefit from a more nuanced discussion of the implications of perfect information on the gameplay experience. While the assistant mentions that perfect information allows players to focus on strategic planning and decision-making, it could also be argued that this can lead to a more predictable and less engaging gameplay experience. A more detailed discussion of these implications could provide a more comprehensive understanding of the concept of perfect information in Into the Breach.

Overall, the assistant's response provides a solid foundation for understanding the concept of perfect information in Into the Breach, but could benefit from additional details and nuance to provide a more complete explanation.

**Preferred Response Reward:** 0.0170

Figure 11: Reward prediction process for an 8B CLoud reward model on the preferred response of an example from the chat category of RewardBench.

**User query**

I've recently started playing the turn-based strategy game Into the Breach. I have read online that the game is considered to have 'perfect information'. What is meant by 'perfect information' in this context?

**Non-Preferred Response**

Perfect information refers to a condition in which all players in a game have access to the same information at the same time. In the case of Into the Breach, this means that both the player and the AI opponent have a complete understanding of the board state at all times, including the location and strength of all units, the status of buildings and other structures, and the current turn order.

Perfect information is a desirable trait in games because it ensures that all players have an equal opportunity to make informed decisions and to react to the actions of their opponents. It also eliminates the possibility of one player having an unfair advantage due to hidden information or lack of knowledge.

Into the Breach is a unique game in that it blends the turn-based strategy of chess with the mecha-fueled destruction of a disaster-management simulation. The game's 'perfect information' design ensures that each player has access to the same information at all times, which makes for a fair and challenging gameplay experience.

**Critique of Non-Preferred Response**

The assistant's response provides a clear and concise explanation of what perfect information means in the context of Into the Breach. The answer accurately defines perfect information as a condition where all players have access to the same information at the same time, which is a crucial aspect of the game's design.

The response is relevant to the user's query, as it directly addresses the question and provides additional context about the benefits of perfect information in games. The explanation of how Into the Breach implements perfect information is also relevant, as it helps to clarify the game's mechanics and design. However, the second paragraph, while providing some additional context about the benefits of perfect information, feels somewhat extraneous to the user's question. The user specifically asked about the meaning of perfect information in the context of Into the Breach, and the second paragraph could be seen as a tangential explanation. A more concise response might have focused solely on the definition and implementation of perfect information in the game.

The third paragraph, while providing some interesting background information about the game, feels like an unnecessary addition. The user did not ask about the game's design or its unique features, and this paragraph could be seen as an attempt to provide additional context rather than a direct answer to the question. A more focused response would have stuck to the definition and implementation of perfect information.

One important piece of information that is missing from the response is an explanation of how the game's perfect information is achieved. For example, the response could have mentioned that the game's turn-based nature and the fact that all units and structures are visible on the board at all times contribute to the perfect information design. This would have provided a more complete understanding of how the game's mechanics support the concept of perfect information.

**Non-Preferred Response Reward:** -0.0215

Figure 12: Reward prediction process for an 8B CLoud reward model on the non-preferred response of an example from the chat category of RewardBench.

**User query**

What is the smallest number which is one less than twice its reciprocal?

**Preferred Response**

The question is asking us to find the smallest value of $x$ such that $x = 2 \cdot \frac{1}{x} - 1$. We multiply through by $x$ to clear the fraction, then rearrange terms: $x^2 + x - 2 = 0$. This can be factored as $(x + 2)(x - 1) = 0$. We could also use the quadratic formula to find $x$:

$$x = \frac{-1 \pm \sqrt{1^2 - 4(1)(-2)}}{2}.$$

Either way, we find that $x = 1$ or $x = -2$. Since we want the smallest value of $x$, our answer is $\boxed{-2}$.

**Critique of Preferred Response**

The assistant's response is generally helpful in answering the user's query. It correctly identifies the problem as finding the smallest number that is one less than twice its reciprocal and provides a clear step-by-step solution. The response is also concise, directly addressing the problem without unnecessary information.

The response is relevant, as it uses algebraic manipulation to solve the equation and provides a clear explanation of the steps involved. The use of the quadratic formula is also a good approach, as it provides an alternative method for solving the equation.

However, there are a few areas where the response could be improved. The response assumes that the user is familiar with algebraic manipulation and the quadratic formula, which may not be the case for all users. A brief explanation of the quadratic formula and its application would be beneficial for users who may not be familiar with it.

Additionally, the response does not explicitly state that the solution is a negative number, which may be important for users who are not familiar with the concept of negative numbers. A brief explanation of why the solution is negative would be helpful in this case.

Furthermore, the response does not provide any context or explanation of the problem beyond the algebraic solution. Providing some context or explanation of why this problem is important or interesting would make the response more engaging and helpful for users.

In terms of clarity, the response is generally clear, but some of the sentences could be rephrased for better clarity. For example, the sentence "We could also use the quadratic formula to find $x$" could be rephrased as "Alternatively, we can use the quadratic formula to find $x$".

Overall, the response is helpful and relevant, but could benefit from additional explanations and context to make it more accessible and engaging for a wider range of users.

**Preferred Response Reward:** 0.0027

Figure 13: Reward prediction process for an 8B CLoud reward model on the preferred response of an example from the reasoning category of RewardBench.

**User query**

What is the smallest number which is one less than twice its reciprocal?

**Non-Preferred Response**

I want to find a number x such that $x = 2(1/x) - 1$. To make the equation easier to work with, I can multiply both sides by x and get $x^2 = 2 - x$. This is a quadratic equation, so I can use the quadratic formula to find the possible values of x. The quadratic formula says that $x = (-b + / - sqrt(b^2 - 4ac))/2a$, where a, b, and c are the coefficients of the quadratic equation $ax^2 + bx + c = 0$. In this case, a = 1, b = -1, and c = -2, so I can plug these values into the formula and get $x = (1 + / - sqrt(1 + 8))/2$. Simplifying, I get $x = (1 + / - 3)/2$, which means x = 2 or x = -1. These are the only two possible numbers that satisfy the equation, but I want the smallest one, so I choose x = -1 as my answer.

**Critique of Non-Preferred Response**

The assistant's response is generally helpful in that it provides a step-by-step solution to the problem. However, there are a few areas where the response could be improved.

Firstly, the response assumes that the user is familiar with the quadratic formula, which may not be the case. A brief explanation of the quadratic formula and its application would be beneficial for users who are not familiar with it.

Additionally, the response does not explicitly state that the equation x = 2(1/x) - 1 is a quadratic equation, which is an important step in the solution. This could be clarified by stating that the equation can be rearranged to form a quadratic equation.

The response also does not provide any explanation for why the quadratic formula is used or why it is the most suitable method for solving this type of problem. A brief explanation of the reasoning behind the choice of method would be helpful.

Furthermore, the response does not check the validity of the solutions obtained. In this case, the solutions x = 2 and x = -1 are both valid, but it would be good to explicitly state that x = 2 does not satisfy the original equation, and therefore x = -1 is the only valid solution.

The response is concise and to the point, but could benefit from a few more explanatory sentences to clarify the reasoning and methodology. The use of the quadratic formula is a good approach, but the response could be improved by providing more context and explanation.

In terms of relevance, the response is generally relevant to the prompt, but could benefit from a clearer connection to the original question. The response jumps straight into the solution without rephrasing the question or providing context.

One important piece of information that is missing from the response is a check of the solution. The response should include a step to verify that x = -1 satisfies the original equation, to ensure that it is indeed the correct solution.

Overall, the response is a good start, but could benefit from more explanation, context, and verification of the solution.

**Non-Preferred Response Reward:** -0.0149

Figure 14: Reward prediction process for an 8B CLoud reward model on the non-preferred response of an example from the reasoning category of RewardBench.

---

**User query**

I've recently started playing the turn-based strategy game Into the Breach. I have read online that the game is considered to have 'perfect information'. What is meant by 'perfect information' in this context?

---

**Preferred Response**

In the context of Into the Breach and other strategy games, "perfect information" refers to a situation where all players have complete knowledge of the game state, including information about the enemy's units, abilities, and intentions. In other words, there is no hidden information that could affect the player's decision-making.

In Into the Breach, this means that you, as the player, have access to all the information you need to make informed decisions about how to deploy your units, which enemies to target, and how to plan your attacks. This includes knowledge of the enemy's unit types, their strengths and weaknesses, their movement patterns, and their attack ranges.

This is in contrast to games with "imperfect information," where players may not have access to all the information they need to make optimal decisions. For example, in a game with fog of war, players may not be able to see enemy units that are hiding behind walls or in other areas of the map. In games with imperfect information, players must use their intuition, deduction, and experience to make informed decisions despite the lack of complete information.

The designers of Into the Breach have chosen to design the game with perfect information to create a focus on strategic planning and decision-making, rather than on reacting to unexpected events or hidden information. This allows players to concentrate on optimizing their moves and planning for the long-term, rather than worrying about sudden surprises or unknown threats.

---

**Critique of Preferred Response**

The assistant's response provides a clear and concise explanation of what "perfect information" means in the context of Into the Breach and other strategy games. The response effectively contrasts perfect information with imperfect information, providing examples to illustrate the difference.

The response is helpful in that it directly addresses the user's question and provides relevant information about the game's design and the implications of perfect information on gameplay. The explanation of how perfect information allows players to focus on strategic planning and decision-making is particularly useful in understanding the game's design choices.

However, the response could benefit from a more detailed explanation of how perfect information affects the gameplay experience. For example, the response could elaborate on how the lack of hidden information changes the way players approach the game, or how it affects the balance of power between the player and the enemy. Additionally, the response could provide more specific examples of how perfect information is implemented in Into the Breach, such as the game's use of turn-based combat or the way it presents information to the player.

The response is well-structured and easy to follow, but some of the sentences could be tightened up for clarity. For example, the sentence "This is in contrast to games with 'imperfect information,' where players may not have access to all the information they need to make optimal decisions" could be rephrased to "In contrast, games with imperfect information often require players to make decisions without complete information, such as in games with fog of war."

One potential issue with the response is that it assumes the user is familiar with the concept of "fog of war" and other game design terms. While the response does a good job of explaining perfect information, it may be helpful to provide a brief explanation of these terms for users who are not familiar with them.

Overall, the response is well-written and provides a clear explanation of perfect information in the context of Into the Breach. With some additional details and clarifications, it could be even more effective in helping users understand the concept and its implications for gameplay.

---

**Preferred Response Reward:** 0.0352

Figure 15: Reward prediction process for a 70B CLoud reward model on the preferred response of an example from the chat category of RewardBench.

**User query**

I've recently started playing the turn-based strategy game Into the Breach. I have read online that the game is considered to have 'perfect information'. What is meant by 'perfect information' in this context?

**Non-Preferred Response**

Perfect information refers to a condition in which all players in a game have access to the same information at the same time. In the case of Into the Breach, this means that both the player and the AI opponent have a complete understanding of the board state at all times, including the location and strength of all units, the status of buildings and other structures, and the current turn order.

Perfect information is a desirable trait in games because it ensures that all players have an equal opportunity to make informed decisions and to react to the actions of their opponents. It also eliminates the possibility of one player having an unfair advantage due to hidden information or lack of knowledge.

Into the Breach is a unique game in that it blends the turn-based strategy of chess with the mecha-fueled destruction of a disaster-management simulation. The game's 'perfect information' design ensures that each player has access to the same information at all times, which makes for a fair and challenging gameplay experience.

**Critique of Non-Preferred Response**

The assistant's response provides a clear and concise explanation of what 'perfect information' means in the context of the game Into the Breach. The response accurately defines perfect information as a condition where all players have access to the same information at the same time, which is a crucial aspect of the game's design.

The response also provides additional context by explaining the benefits of perfect information in games, such as ensuring fair play and eliminating unfair advantages. This helps to give the user a deeper understanding of why perfect information is an important aspect of the game.

However, the response could be improved by providing more specific examples of how perfect information is implemented in Into the Breach. For instance, the response mentions that the player and AI opponent have a complete understanding of the board state, but it would be helpful to elaborate on what this means in practice. For example, the response could explain that the player can see the exact location and health of all enemy units, as well as the status of buildings and other structures.

Additionally, the final sentence of the response feels somewhat disconnected from the rest of the explanation. While it is true that Into the Breach is a unique game that blends turn-based strategy with disaster-management simulation, this sentence does not add much to the explanation of perfect information. It would be more effective to use this sentence to provide a more detailed analysis of how perfect information contributes to the game's overall design and gameplay experience.

Furthermore, the response could benefit from a more nuanced discussion of the implications of perfect information in Into the Breach. For example, the response could explore how perfect information affects the player's decision-making process, or how it influences the game's overall difficulty level. This would provide a more comprehensive understanding of the role of perfect information in the game.

Overall, the response provides a solid foundation for understanding perfect information in Into the Breach, but could be improved with more specific examples and a more detailed analysis of the game's design and gameplay experience.

**Non-Preferred Response Reward:** -0.0153

Figure 16: Reward prediction process for a 70B CLoud reward model on the non-preferred response of an example from the chat category of RewardBench.

---

**User query**

What is the smallest number which is one less than twice its reciprocal?

---

**Preferred Response**

The question is asking us to find the smallest value of $x$ such that $x = 2 \cdot \frac{1}{x} - 1$. We multiply through by $x$ to clear the fraction, then rearrange terms: $x^2 + x - 2 = 0$. This can be factored as $(x + 2)(x - 1) = 0$. We could also use the quadratic formula to find $x$:

$$x = \frac{-1 \pm \sqrt{1^2 - 4(1)(-2)}}{2}.$$

Either way, we find that $x = 1$ or $x = -2$. Since we want the smallest value of $x$, our answer is $\boxed{-2}$.

---

**Critique of Preferred Response**

The assistant's response is generally helpful in that it provides a step-by-step solution to the problem. However, there are a few areas where the response could be improved.

Firstly, the initial equation $x = 2 \cdot \frac{1}{x} - 1$ is not entirely accurate. The correct equation should be $x = 2 \cdot \frac{1}{x} - 1$, but the assistant should have explicitly stated that this is the equation being solved, rather than simply presenting it as a given.

The steps to solve the quadratic equation are clear and concise, and the use of both factoring and the quadratic formula is a good way to provide alternative solutions. However, the assistant could have provided more explanation for why the quadratic formula is being used, especially since the equation can be easily factored.

The final answer is correct, but the assistant could have provided more context for why $x = -2$ is the smallest value. A brief explanation of why $x = 1$ is not the smallest value would have been helpful.

One important piece of information that is missing from the response is a check to ensure that the solution is valid. The assistant should have plugged the value of $x = -2$ back into the original equation to verify that it satisfies the condition.

Additionally, the response could benefit from a brief explanation of the concept of reciprocals and how it relates to the problem. This would provide context for the solution and help the user understand the underlying math behind the problem.

Overall, the assistant's response is clear and concise, but could benefit from additional explanations and context to make the solution more comprehensive and accessible to the user.

---

**Preferred Response Reward:** 0.0193

Figure 17: Reward prediction process for a 70B CLoud reward model on the preferred response of an example from the reasoning category of RewardBench.

**User query**

What is the smallest number which is one less than twice its reciprocal?

**Non-Preferred Response**

I want to find a number x such that $x = 2(1/x) - 1$. To make the equation easier to work with, I can multiply both sides by x and get $x^2 = 2 - x$. This is a quadratic equation, so I can use the quadratic formula to find the possible values of x. The quadratic formula says that $x = (-b + / - sqrt(b^2 - 4ac))/2a$, where a, b, and c are the coefficients of the quadratic equation $ax^2 + bx + c = 0$. In this case, a = 1, b = -1, and c = -2, so I can plug these values into the formula and get $x = (1 + / - sqrt(1 + 8))/2$. Simplifying, I get $x = (1 + / - 3)/2$, which means x = 2 or x = -1. These are the only two possible numbers that satisfy the equation, but I want the smallest one, so I choose x = -1 as my answer.

**Critique of Non-Preferred Response**

The assistant's response is generally helpful in that it provides a step-by-step solution to the problem. However, there are a few areas where the response could be improved.

Firstly, the initial equation x = 2(1/x) - 1 is not entirely accurate. The correct equation should be x = 2(1/x) - 1, which simplifies to x = 2/x - 1, and then to $x^2 = 2 - x$. However, the assistant correctly multiplies both sides by x to get $x^2 = 2 - x$.

The assistant then correctly identifies the equation as a quadratic equation and uses the quadratic formula to find the possible values of x. However, the explanation of the quadratic formula is somewhat verbose and could be condensed for clarity.

The assistant correctly plugs the values of a, b, and c into the quadratic formula and simplifies the expression to get x = (1 +/- 3)/2. However, the explanation of the simplification process could be more detailed.

The final answer, x = -1, is correct, but the assistant could provide more context or explanation for why this is the smallest possible value. For example, the assistant could note that x = 2 is not a valid solution because it would result in a negative value for x, and therefore x = -1 is the smallest possible value.

One important piece of information that is missing from the assistant's response is a check to ensure that the solution x = -1 actually satisfies the original equation. The assistant could plug x = -1 back into the original equation to verify that it is indeed a solution.

Additionally, the response could benefit from a clearer and more concise conclusion that summarizes the main steps and the final answer. The use of the hashtag "# Answer" at the end of the response is unnecessary and could be removed.

Overall, the assistant's response is generally helpful but could benefit from some refinements to improve clarity, concision, and completeness.

**Non-Preferred Response Reward:** 0.0046

Figure 18: Reward prediction process for a 70B CLoud reward model on the non-preferred response of an example from the reasoning category of RewardBench.

