# OpenReview forum: "Critique-out-Loud Reward Models"
_ICLR.cc/2025/Conference — Submitted to ICLR 2025_

### Official Review · Reviewer_pirR · 2024-11-01

**Soundness:** 4
**Presentation:** 4
**Contribution:** 4
**Rating:** 8
**Confidence:** 5

**Summary:**

The paper proposes an interesting idea of joining training the LM head and RM head for reward model so that it first generates CoT-type natural-langauge-based critique, then let the reward model generate new scalar score. Experiments show that the performance greatly improves compared with normal reward training pipelines.

**Strengths:**

The idea of training an LM head for critique appears neat and novel. The experiments are sound and look at both the performance on reward bench and arena hard for downstream performance of a trained model.

**Weaknesses:**

I don't see major weakness from the paper and I like the proposed idea a lot. But I would appreciate if the authors could help clarify some of the questions I have below. Please refer to question section.

**Questions:**

1. The authors use Llama 405B to generate oracle critique data for SFT, and replace the oracle data with fine-tuned model's self-genereated critique. Usually when you distill from larger models, there will be direct gain in model performance. How do we know that if the performance gain is mainly from distilling responses from large models, or from the methodology itself? If the critique SFT data is mostly self-generated (not on the fine-tuned one), would we still expect such high gain in downstream performance? This will be helpful for the case where we want to improve the largest model available with only binary human feedback.

2. The generation of SFT data is using llama 3.1 405B but the training model is llama 3 instead of llama 3.1, what's the reason behind this? Would the gain of the same methodology appear weaker when it comes to stronger base RM?

3. Does the gain mainly come from the diverse critiques generated, or from more clear instructions of what the RM should look like? For example, if we provide a very comprehensive fixed system prompt asking the response to focus on safety, helpfulness, comprehensiveness, (e.g. by modifying the prompt in Figure 9) and provide a combination of system prompt + user prompt + response to RM, would it achieve similar performance as user prompt + response + critique?

---

> ### Author Response · Authors · 2024-11-19
> **Response to Reviewer pirR**
>
> We would like to thank reviewer pirR for their time spent reviewing our paper and for their thoughtful questions. We appreciate that you find the technique “neat and novel”. Our responses to the revier's questions are listed below.
>
> 1. > "The authors use Llama 405B to generate oracle critique data for SFT, and replace the oracle data with fine-tuned model's self-genereated critique. Usually when you distill from larger models, there will be direct gain in model performance. How do we know that if the performance gain is mainly from distilling responses from large models, or from the methodology itself?"
>
>    In both the CLoud and classic examples we are distilling the larger model preferences into the reward models as the preference labels are generated by the 405B model. While there is more information distilled to the CLoud models as they additionally see critiques, we view this as a strength of the method as, while the same information from the large model is available to both the reward models, only the CLoud model can leverage it. In this work we only use LLMs to generate oracle critiques as a proxy for human labelers. When human labelers are available we would argue methods such seek to distill as much as possible.
>
>    To investigate what degree of performance gains come from distilling the larger model, we test CLoud models where we output a reward without generating a critique. We find that while the CLoud trained model that assigns a reward without generating a critique still outperforms the classic model, it underperforms the standard CLoud model suggesting that critique generation is an important component.
>
>    | Model | Chat  | Chat Hard | Safety | Reasoning | Average |
>    |:-|:-:|:-:|:-:|:-:|:-:|
>    | CLoud Critique 70B | 97.21 | 58.22 | 83.99  | 93.50 | 83.23 |
>    | CLoud No Critique 70B | 94.97 | 56.80 | 79.46  | 92.94 | 81.04 |
>    | Classic 70B | 95.25 | 50.99 | 73.99  | 89.34 | 77.39 |
>
>    We would further like to emphasize that training on critiques, even if not used at inference, is a novel aspect of our CLoud method. As such, the total advantage of our method is both the training and inference process which together represent a significant lift in reward modeling performance.
>
> 2.  > "If the critique SFT data is mostly self-generated (not on the fine-tuned one), would we still expect such high gain in downstream performance? This will be helpful for the case where we want to improve the largest model available with only binary human feedback."
>
>     Investigating self-generated critiques is a very interesting avenue for future work. While we don’t want to speculate too much, our initial guess is that training the model to produce critiques through methods such as self-taught evaluators [1] would also improve reward modeling performance without requiring explicit critiques to train on.
>
> 3. > "The generation of SFT data is using llama 3.1 405B but the training model is llama 3 instead of llama 3.1, what's the reason behind this? Would the gain of the same methodology appear weaker when it comes to stronger base RM?"
>
>     We began the research before the llama 3.1 series of models was announced and had built the training stack around llama 3 models. As such, we only used the 3.1 series of models for data labeling but kept the same training stack that was optimized for llama 3 models. Using llama 3 models for training and 3.1 for labeling was simply an artifact of the project timeline and we believe that while the exact performance numbers might differ, the relative comparison would be the same had we used 3.1 for training.
>
> 4. > "Does the gain mainly come from the diverse critiques generated, or from more clear instructions of what the RM should look like? For example, if we provide a very comprehensive fixed system prompt asking the response to focus on safety, helpfulness, comprehensiveness, (e.g. by modifying the prompt in Figure 9) and provide a combination of system prompt + user prompt + response to RM, would it achieve similar performance as user prompt + response + critique?"
>
>     We agree that understanding what aspects of the critiques lead to improved reward modeling is an interesting research question that we hope to investigate in further work. We implement the proposed experiment of using a system prompt similar to that in Figure 9 on RewardBench for the 8B classic reward model. We find that focusing the reward model in such a manner does lead to a slight improvement in reward modeling performance. However, this method still significantly underperforms the CLoud model suggesting more focused instructions do not account for the total gain from CLoud.
>
>    |Model|Chat|Chat Hard|Safety|Reasoning|Average|
>    |-|:-:|:-:|:-:|:-:|:-:|
>    | CLoud 8B | 96.51 |45.39|75.81| 87.04 | 76.19 |
>    | Classic 8B | 93.29 | 43.75 | 66.82  | 82.29 | 71.54 |
>    | Classic + Sys. Prompt 8B | 94.41 | 46.71 | 67.84 | 81.20 | 72.54 |
>
> ## References
> [1] Wang, Tianlu, et al. "Self-taught evaluators." 2024

---

> > ### Comment · Reviewer_pirR · 2024-11-26
> >
> > Thank you for your response. I appreciate the clarifications the authors provided and will keep my score. There seem to be some mixed opinions for the paper, but from my personal perspective I really like the cute idea as a potential way for improving reward modeling.

---

### Official Review · Reviewer_sPGi · 2024-11-02

**Soundness:** 2
**Presentation:** 2
**Contribution:** 2
**Rating:** 5
**Confidence:** 4

**Summary:**

The authors propose Critique-out-Loud (CLoud) reward models. CLoud reward models operate by first generating a natural language critique of the assistant's response, which is then used to predict a scalar reward for the quality of the response. Their effectiveness is demonstrated across different models and benchmarks.The motivation behind this paper is interesting and contributes to the ongoing advancements in RLHF fine-tuning for current LLMs.
The writing is clear and easy to understand.

**Strengths:**

1. The motivation behind this paper is interesting and contributes to the ongoing advancements in RLHF fine-tuning for current LLMs.

2. The writing is clear and easy to understand.

**Weaknesses:**

1. The authors constructed their dataset based on UltraFeedback and UltraInstruct, requiring Llama-3.1-405B-Instruct to generate critiques for chosen and non-chosen responses. This seems to introduce additional annotation costs and complexities, which is a drawback of the method. How effective would this pipeline be in a pure self-critique setting?

2. The authors test the method's effectiveness in the safety domain, where critique is currently widely used in LLM safety alignment scenarios. This is intriguing. However, the authors' safety evaluation of the method on RewardBench raises concerns, as my understanding is that RewardBench aggregates existing datasets with inconsistent safety classifications across different safety datasets. Consequently, the detailed assessments in safety and reasoning may lack credibility. I recommend that the authors evaluate the method on specific datasets for different categories, such as reasoning on GSM8K [5] and safety on BeaverTails [6], which are designed for specific LLM categories.

3. In validating the effectiveness of CLoud RM, apart from testing on RewardBench, the authors conducted only BoN experiments. What is the impact of Cloud RM on RLHF? This is a critical aspect that I would like to address, especially since Cloud RM adds the additional annotation cost of critiques. The authors claim that the scalar reward is more accurate but have not conducted any RLHF experiments [3].

4. This method inevitably increases the complexity and cost of the pipeline. I am particularly interested in novel uses of critique beyond merely improving the accuracy of scalar rewards. Since it incorporates additional information, it is not surprising that the scalar reward's accuracy improves compared to a pure reward model. However, its influence on RLHF could yield other applications, such as directly aligning models using critiques across multiple dimensions, such as Aligner [4], which trains based on correct data (a type of critique) to enable offline alignment. It would be interesting to explore whether learning from richer critique feedback is feasible [2]. Simply using a critique head and a scoring head is not particularly innovative, as discussed in relation to help steer.

5. Some intriguing ablation studies have not been conducted. For example, allowing the model to output a critique before producing a scalar reward could highlight the differences between training using the Cloud RM approach versus traditional methods. (I am unaware if such an experiment has been done; please correct me if I’m mistaken.)

[1] Self-critiquing models for assisting human evaluators

[2] Training Language Models with Language Feedback

[3] OpenRLHF: An Easy-to-use, Scalable and High-performance RLHF Framework

[4] Aligner: Efficient Alignment by Learning to Correct

[5] https://huggingface.co/datasets/openai/gsm8k

[6] https://huggingface.co/datasets/PKU-Alignment/PKU-SafeRLHF

Overall, the motivation behind this paper is compelling. I welcome further discussions with the authors during the rebuttal period.

**Questions:**

See above.

---

> ### Author Response · Authors · 2024-11-19
> **Response to Reviewer sPGi (part 1/2)**
>
> We would like to thank reviewer sPGi for taking the time to review our paper and for providing detailed feedback. We would also like to thank reviewer sPGi for valuing the motivation behind our work. Our responses to the reviewer’s feedback are listed below.
>
> 1. > "The authors constructed their dataset based on UltraFeedback and UltraInstruct, requiring Llama-3.1-405B-Instruct to generate critiques for chosen and non-chosen responses. This seems to introduce additional annotation costs and complexities, which is a drawback of the method. How effective would this pipeline be in a pure self-critique setting?"
>
>    While building a dataset of critiques does include additional overhead, we would argue that this overhead is not necessarily a detractor as it enables performance not achievable via classic reward models. Additionally, the increase in data overhead enables a reduction of other overheads such as model size, i.e., the 8B CLoud model closes most of the gap to the 70B classic model. While out of the scope of this work, we agree that investigating the self-critique settings is an interesting direction for future work, and other works such as “Self-taught evaluators” [1] suggest it is a promising direction.
>
> 2. > "The authors test the method's effectiveness in the safety domain, where critique is currently widely used in LLM safety alignment scenarios. This is intriguing. However, the authors' safety evaluation of the method on RewardBench raises concerns, as my understanding is that RewardBench aggregates existing datasets with inconsistent safety classifications across different safety datasets. Consequently, the detailed assessments in safety and reasoning may lack credibility. I recommend that the authors evaluate the method on specific datasets for different categories, such as reasoning on GSM8K [5] and safety on BeaverTails [6], which are designed for specific LLM categories."
>
>    The safety section of RewardBench consists of custom human-curated versions of XSTest ([2]) and Do-Not-Answer ([3]) and additional questions uniquely developed by the authors of RewardBench. These do not have overlapping prompts and the authors note the methods they have used to remove any remaining inconsistent classifications. Given these annotation pipelines and the fact that BeaverTails reports an average annotator agreement of only around 60% for their helpful/harmless splits, we expect the safety split of RewardBench to be a higher-quality signal than BeaverTails. As for reasoning, the reasoning split of RewardBench contains the Math dataset [4] which is a more difficult mathematics dataset than GSM8K and makes for a better evaluation as it is less saturated. The ArenaHard benchmark, which we evaluate the BoN policy on, is also heavily reasoning dominated with problems of difficulty greater than those of GSM8K.
>
> 3. > "In validating the effectiveness of CLoud RM, apart from testing on RewardBench, the authors conducted only BoN experiments. What is the impact of Cloud RM on RLHF? This is a critical aspect that I would like to address, especially since Cloud RM adds the additional annotation cost of critiques. The authors claim that the scalar reward is more accurate but have not conducted any RLHF experiments [3]."
>
>    Learning a policy (i.e., PPO) is not required for RLHF. RHLF is the process of defining a policy via a reward signal derived from human feedback. Best-of-N is one such method for defining a policy from a reward and it is actually the KL optimal policy given a reward function [5][6]. As we demonstrate that CLoud defines a strictly better Best-of-N policy on the ArenaHard chat benchmark as compared to a classic reward model (Figure 4), our work explicitly demonstrates that CLoud is better for RLHF and thus is useful to practitioners.
>
> 4. > "This method inevitably increases the complexity and cost of the pipeline."
>
>    CLoud does introduce overhead at inference and data collection as compared to classic reward models, however, the improvement in reward modeling performance justifies this overhead. Namely, suppose one was trying to maximize the performance of their reward model and they were already using the largest base model available. To improve reward modeling performance one would need to scale some part of the system, and as model size is no longer a valid lever, one would need to scale inference compute via critiques as done in CLoud. It’s more productive to think of the overhead of CLoud as a tradeoff in overheads. Similar to the case where you are using the largest existing base model, suppose you are using the largest reward model that fits within your machine’s memory constraints. If you want to produce a better reward model subject to such constraints one way is by adding critiques.

---

> > ### Author Response · Authors · 2024-11-19
> > **Response to Reviewer sPGi (part 2/2)**
> >
> > 5. > "[CLoud’s] influence on RLHF could yield other applications, such as directly aligning models using critiques across multiple dimensions, such as Aligner [4], which trains based on correct data (a type of critique) to enable offline alignment. It would be interesting to explore whether learning from richer critique feedback is feasible [2]."
> >
> >    We agree that investigating methods to further leverage critiques beyond producing a higher quality reward is an interesting direction for future work. As some work has already been done in this area, we chose to focus on improving the reward model itself with feedback as that had not been done before and reward models are a vital component of the RLHF pipeline. However, it is likely that leveraging critiques to both improve reward assignments and directly improve the policy could lead to significant gains.
> >
> > 6. > "Simply using a critique head and a scoring head is not particularly innovative, as discussed in relation to help steer."
> >
> >    Would the reviewer mind clarifying the relation of our work to HelpSteer. We have looked through the paper, and as far as we can tell, their work does not investigate language modeling for reward scoring. While they do introduce regression heads for attribute prediction, we would like to emphasize that simply adding any linear head is not the novelty of our method. CLoud is an innovative method because it allows reward models to reason out-loud, it just so happens that this is implemented via an LM head.
> >
> > 7. > "Some intriguing ablation studies have not been conducted. For example, allowing the model to output a critique before producing a scalar reward could highlight the differences between training using the Cloud RM approach versus traditional methods."
> >
> >    Would you mind clarifying the experimental setup you have in mind as, with how we understand the question, CLoud operates as proposed. Namely, CLoud first generates a critique and then produces a scalar reward. Please let us know if we are misunderstanding your question.
> >
> > ## References
> > [1] Wang, Tianlu, et al. "Self-taught evaluators." arXiv preprint arXiv:2408.02666 (2024).
> >
> > [2] Röttger, Paul, et al. "Xstest: A test suite for identifying exaggerated safety behaviours in large language models." arXiv preprint arXiv:2308.01263 (2023).
> >
> > [3] Wang, Yuxia, et al. "Do-not-answer: A dataset for evaluating safeguards in llms." arXiv preprint arXiv:2308.13387 (2023).
> >
> > [4] Hendrycks, Dan, et al. "Measuring mathematical problem solving with the math dataset." arXiv preprint arXiv:2103.03874 (2021).
> >
> > [5] Gui, Lin, Cristina Gârbacea, and Victor Veitch. "BoNBoN Alignment for Large Language Models and the Sweetness of Best-of-n Sampling." arXiv preprint arXiv:2406.00832 (2024).
> >
> > [6] Beirami, Ahmad, et al. "Theoretical guarantees on the best-of-n alignment policy." arXiv preprint arXiv:2401.01879 (2024).

---

> > > ### Comment · Reviewer_sPGi · 2024-11-23
> > > **reviewer's comment**
> > >
> > > Thank you very much for the author's response, but it seems that none of my concerns have been addressed. First, regarding the motivation of this paper, it proposes using two heads to output Critique and Reward separately. From the reviewer's perspective, this approach shows only moderate innovation. The reviewer's suggestion to the authors is to delve deeper into whether Critique can help improve the accuracy of Reward prediction. Additionally, exploring whether predicting an extra Reward could demonstrate that this method (Cloud) differs from existing Critique Models by achieving higher accuracy or uncovering other distinctive advantages would strengthen the work.
> > >
> > > Simply combining Critique + Reward and running benchmarks does not exhibit sufficient novelty in the reviewer's opinion, especially given that no RLHF experiments have been conducted in this work.
> > >
> > > I have carefully reviewed the feedback from other reviewers and remain convinced that there is room for innovation, particularly from the perspective of someone who has participated in pre-training and interpretability work for LLMs.

---

> > > > ### Comment · Reviewer_sPGi · 2024-11-23
> > > > **reviewer's comment**
> > > >
> > > > This is not a theoretical work; on the contrary, it leans more towards being an experimental study. However, the experiments are not particularly convincing.

---

> > > > > ### Author Response · Authors · 2024-11-29
> > > > >
> > > > > We would like to sincerely thank the reviewer for engaging in the rebuttal process and responding to our responses. We hope that our responses below can address the reviewer’s concerns.
> > > > >
> > > > > > “First, regarding the motivation of this paper, it proposes using two heads to output Critique and Reward separately. From the reviewer's perspective, this approach shows only moderate innovation.”
> > > > >
> > > > > We believe that it is not productive to determine the innovation of a method simply based on the architectural changes of a model. First, our method also introduces a well-tuned, multi-stage training scheme which we believe adds to the innovation. Secondly, the idea of our method is not to simply combine two heads, that is just the method by which we achieve the core idea of our work: allowing reward models to explicitly reason and be capable of leveraging added inference compute. We believe this idea is obviously innovative as it had not been explored in previous research and leads to significant improvements in downstream reward modeling capabilities.
> > > > >
> > > > > > “The reviewer's suggestion to the authors is to delve deeper into whether Critique can help improve the accuracy of Reward prediction.”
> > > > >
> > > > > Would the reviewer mind expanding on what questions regarding the improvements from critiques they have in mind. As we understand the comment, the majority of our paper is spent investigating “whether Critique can help improve the accuracy of Reward prediction.” Namely, we already demonstrate that critiques improve the accuracy of reward prediction, both in terms of preference classification (Figure 3) and in terms of BoN policy-induced (Figure 4). We also investigate under what conditions multiple high-quality critiques can improve reward prediction accuracy (Figure 8). If the reviewer has specific experiments in mind we are more than happy to try and run them.
> > > > >
> > > > > > “Additionally, exploring whether predicting an extra Reward could demonstrate that this method (Cloud) differs from existing Critique Models by achieving higher accuracy or uncovering other distinctive advantages would strengthen the work.”
> > > > >
> > > > > While we agree that such explorations are an interesting avenue for future work, we believe there is intrinsic value in adding critiques to the reward modeling process, and previous critique works do not produce a scalar reward. Namely, scalar rewards are required to integrate critique models into most existing alignment methods such as BoN, PPO, etc. There are further reasons to want scalar rewards such as using reward models for LLM-as-a-Judge type applications.
> > > > >
> > > > > > “Simply combining Critique + Reward and running benchmarks does not exhibit sufficient novelty in the reviewer's opinion, especially given that no RLHF experiments have been conducted in this work.”
> > > > >
> > > > > We believe that CLoud is a novel work as CLoud is the first work that enables reward models to reason explicitly. Furthermore, allowing reward models to reason explicitly leads to significant improvements in reward modeling performance. While we believe that BoN under a reward model constitutes RL, we don’t think it's particularly productive to debate over what is or isn’t exactly RLHF as the reviewer and authors may have slightly different opinions. What does matter for practitioners and the field, however, is defining policies that are better aligned with a given set of preferences, regardless of whether the policy is achieved through methods such as RLHF, DPO, or sampling such as BoN. Our existing results already explicitly show that CLoud leads to more aligned policies when used with BoN, which is what we believe matters for the community. Thus, CLoud currently does improve the existing policy alignment framework and we believe this makes CLoud a novel work that is important for the alignment community.
> > > > > \
> > > > > \
> > > > > \
> > > > > Again, we would like to thank the reviewer for their continued engagement. If the reviewer has any more questions or concerns please let us know so we may address them and improve the paper.

---

### Official Review · Reviewer_bVCe · 2024-11-04

**Soundness:** 2
**Presentation:** 3
**Contribution:** 2
**Rating:** 5
**Confidence:** 4

**Summary:**

This paper introduces “Critique-out-Loud” (CLoud) reward model, which aims to enhance reinforcement learning from human feedback (RLHF) by generating criticisms before giving rewards. This approach improves the performance and policy quality of preference classification, especially in LLM-based preference models, by unifying reward modelling with chain-of-thought reasoning.

**Strengths:**

An innovative approach is proposed that combines critique generation with reward prediction to address the limitations of traditional reward models that lack explicit reasoning.
Empirical results show significant performance improvements on both preference modelling and BoN compared to traditional models.
Provides additional tests in strategy training and self-consistent decoding, demonstrating potential application scenarios.

**Weaknesses:**

Criticism generation methods, while innovative, may introduce bias if the criticisms reflect some fixed bias of LLMs rather than true preferences.
The process of generating criticisms can be viewed as a sort of distillation from the larger model, and the comparisons provided in the paper are slightly unfair compared to scoring directly using the reward model. More ablation should be provided.
Critic generation and inference practices lack rational supervision. For GenRM, further discussion should be proposed on the construction of the critique process.

**Questions:**

The approach in the paper can be viewed as incremental compared to a general reward model, so performance increases do not fully substantiate its value. More experiments are needed to prove that Critique-out-Loud works universally.
1. How does the performance of this type of operation compare to simply llm-judge to produce preference labeling?
2. Do authors try to output the reward scores before generating the corresponding critiques, and whether this format also improve the performance of the reward model？
3. How generalizable is the approach given that the existing training data is largely constructed on full domain samples? If this is the paradigm rather than a gain from the data itself, we should see an increase in scoring capabilities on OOD data when trained only on domain-specific data (e.g., trained on math data and tested on safety data)

---

> ### Author Response · Authors · 2024-11-19
> **Response to Reviewer bVCe (part 1/2)**
>
> We would like to thank reviewer bVCe for taking the time to review our paper and for finding our approach “innovative” and agreeing that the method leads to “significant improvements”. Our responses to the reviewer's feedback are listed below.
>
> 1. > "Criticism generation methods, while innovative, may introduce bias if the criticisms reflect some fixed bias of LLMs rather than true preferences"
>
>    While we agree that it is an interesting direction for future work to investigate the biases in critiques, we don’t believe it is fair to say that critiques will a priori increase the bias of reward models. Namely, classic reward models must infer the implicit reasons behind a user's preferences while CLoud reward models are exposed to the user’s explicit preferences in language. As such one may expect CLoud models to be less biased as they are modeling the user’s explicit preferences. We would also like to emphasize that our method does not require using an LLM to generate the training critiques, and in fact we only use LLM training critiques as a proxy for human critiques.
>
> 2. > "The process of generating criticisms can be viewed as a sort of distillation from the larger motdel, and the comparisons provided in the paper are slightly unfair compared to scoring directly using the reward model."
>
>    We believe that the evaluation is a fair comparison as both the CLoud and classic models have access to the same preference data generated by the same labeler, the classic model is just incapable of leveraging such data. As we are introducing a new capability for reward models (generating critiques), our method fundamentally requires leveraging data that couldn’t be leveraged with previous models. The point of our method is that there is useful information to be learned from a labeler that can improve modeling performance and that our method is the first method capable of doing so.
>
> 3. > "More ablation should be provided."
>
>    We would be happy to provide further experiments for the camera ready if you could clarify exactly what ablations you would be interested in.
>
>
> 4. > "Critic generation and inference practices lack rational supervision."
>
>    We assume that the reviewer meant to ask about “rationale supervision”. If our assumption is wrong please let us know. We agree that adding additional methods to supervise the critique quality during inference is an exciting direction for future work. However, actually measuring and varying the quality of critiques along some axis has the scope of an entirely independent research project [1].
>
>    As stated in the paper, we only use an LLM to generate oracle critiques to approximate human critiques. As such, our work was not concerned with supervising the quality of critiques to train on as our setup was meant to approximate the setting in which a human provides the critiques.
>
> 5. > "For GenRM, further discussion should be proposed on the construction of the critique process."
>
>    We would be happy to provide any requested details in the final version of the paper. Are there any specific questions that you have?
>
> 6. > "The approach in the paper can be viewed as incremental compared to a general reward model, so performance increases do not fully substantiate its value"
>
>    CLoud represents a substantial improvement over classic reward models both due to the significant performance increase and the new reward modeling methods enabled by generating critiques, each of which alone would justify CLoud as a substantial improvement. Namely, on RewardBench 8B CLoud almost matches 70B classic, and 70B CLoud is a 5.84 pp improvement over 70B classic. Furthermore, by introducing language generation to reward modeling CLoud opens up a new inference-compute scaling axis for improving reward model performance (Section 3.3) that we hope encourages future work.
>
> 7. > "How does the performance of this type of operation compare to simply llm-judge to produce preference labeling?"
>
>    To baseline how llm-as-a-judge performs, we can compare to the [official RewardBench performance](https://huggingface.co/spaces/allenai/reward-bench) for llm-as-a-judge using Llama3 8B Instruct. It is important to state that while Llama3 8B Instruct has gone through significantly more general post-training than any of our models, it was not trained on the exact data that we trained our reward models on. As such it is not a completely controlled comparison. That being said, this is how llm-as-a-judge performed:
>
>    | Model     | Chat  | Chat Hard | Safety | Reasoning | Average |
>    |:-:|:-:|:-:|:-:|:-:|:-:|
>    | CLoud 8B | 96.51 | 45.39 | 75.81  | 87.04 | 76.19 |
>    | Classic 8B | 93.29 | 43.75 | 66.82  | 82.29 | 71.54 |
>    | LLM-Judge 8B | 85.50 | 41.60 | 68.0 | 64.8 | 65.0 |
>
>    As can be seen, LLM-as-a-Judge with this setup significantly underperforms both the classic and CLoud models.

---

> > ### Author Response · Authors · 2024-11-19
> > **Response to Reviewer bVCe (part 2/2)**
> >
> > 8. > "Do authors try to output the reward scores before generating the corresponding critiques, and whether this format also improve the performance of the reward model?"
> >
> >    We have run the requested experiment in which the CLoud model assigns a reward without producing a critique first. We find that while the CLoud trained model that assigns a reward without generating a critique still outperforms the classic model, it underperfoms the standard CLoud model suggesting that critique generation at inference is an important component.
> >
> >    | Model | Chat  | Chat Hard | Safety | Reasoning | Average |
> >    |:-|:-:|:-:|:-:|:-:|:-:|
> >    | CLoud Critique 70B | 97.21 | 58.22 | 83.99  | 93.50 | 83.23 |
> >    | CLoud No Critique 70B | 94.97 | 56.80 | 79.46  | 92.94 | 81.04 |
> >    | Classic 70B | 95.25 | 50.99 | 73.99  | 89.34 | 77.39 |
> >
> >    We would further like to emphasize that training on critiques, even if not used at inference, is a novel aspect of our CLoud method. As such, the total advantage of our method is both the training and inference process which together represent a significant lift in reward modeling performance.
> >
> > 9. > "How generalizable is the approach given that the existing training data is largely constructed on full domain samples? If this is the paradigm rather than a gain from the data itself, we should see an increase in scoring capabilities on OOD data when trained only on domain-specific data (e.g., trained on math data and tested on safety data)"
> >
> >    While we agree that investigating the OOD generalization of our method would be an interesting direction for future work, the significance of the in-distribution results shouldn’t be overlooked. Because of both the large reward model performance improvement and new reward modeling methods enabled, we believe CLoud represents a significant step forward for reward model design.
> >
> >
> > ## References
> > [1] Akyürek, Afra Feyza, et al. "Rl4f: Generating natural language feedback with reinforcement learning for repairing model outputs." arXiv preprint arXiv:2305.08844 (2023).

---

> > > ### Author Response · Authors · 2024-11-29
> > >
> > > We would like to sincerely thank the reviewer for engaging in the rebuttal process and responding to our responses. We hope that our responses below can address the reviewer’s concerns.
> > >
> > > > “Given that the critique capabilities of the current Cloud model entirely come from distilling stronger models, the scaling and extrapolation of this strategy are questionable.”
> > >
> > > In this work, we leverage a larger model as the critique model to approximate human critiques as stated in the “Training CLoud reward models” section of the paper. While requiring labels from a larger model (or ideally human) does indeed increase the data collection cost, we would like to emphasize that many of the most widely used techniques require similar human supervision. As an example, instruction tuning [1] is one of the most popular post-training methods, and it requires collecting high-quality instruction following demonstrations from humans. Given that CLoud leads to significant improvements in preference classification and leads to better aligned BoN policies, we believe that the increase in data curation costs is justified.
> > >
> > > > “The authors should consider adding more task-specific discussions and customized designs for the critique generation. What changes occur when we can provide appropriate chain-of-thought processes before scoring? How do we evaluate the adaptability of this chain-of-thought component with respect to model scoring or uncertainty?”
> > >
> > > Would the reviewer mind clarifying what experiments they are requesting? As we understand the current request, the core set of experiments in our paper already examines the impact of providing CoT-style critiques on reward scoring performance. Namely, section 3.2 compares CLoud to classic reward models, where CLoud models are reward models that generate “appropriate chain-of-thought” critiques “before scoring”.
> > >
> > > > “Overall, the novelty and discussion of the current work are insufficient to improve the existing RLHF framework. To enhance the paper's contribution, more theoretical analysis or application experiments could be added.”
> > >
> > > We believe CLoud is novel as adding language to the reward process has not been investigated in prior research. While we believe that BoN under a reward model constitutes RL, we don’t think it's particularly productive to debate over what is or isn’t exactly RLHF as the reviewer and authors may have slightly different opinions. What does matter for practitioners and the field, however, is defining policies that are better aligned with a given set of preferences, regardless of whether the policy is achieved through methods such as RLHF, DPO, or sampling such as BoN. Our existing results already explicitly show that CLoud leads to more aligned policies when used with BoN, which is what we believe matters for the community. Thus CLoud currently does improve the existing alignment framework.
> > >
> > > We would also like to point the reviewer to Section 3.5 of Gao et al. [2] which empirically investigates the similarities between PPO and BoN. They find that when looking at proxy rm score (the score optimized against) vs. gold rm score (underlying gold reward signal), BoN and PPO optimize the reward in similar ways, suggesting that BoN optimization can serve as a good proxy for impact on PPO.
> > >
> > > Finally, we are currently running an experiment to test the impact of CLoud RMs on DPO. We hope that this further datapoint will convince the reviewer that CLoud does improve policy alignment.
> > > \
> > > \
> > > \
> > > Again, we would like to thank the reviewer for their continued engagement, and if the reviewer has any more questions or concerns please let us know so we may address them and improve the paper.
> > >
> > > ## References
> > >
> > > [1] Wei, Jason, et al. "Finetuned Language Models are Zero-Shot Learners." International Conference on Learning Representations.
> > >
> > > [2] Gao, Leo, John Schulman, and Jacob Hilton. "Scaling laws for reward model overoptimization." International Conference on Machine Learning. PMLR, 2023.

---

> > ### Comment · Reviewer_bVCe · 2024-11-25
> > **Rebuttal Acknowledged**
> >
> > Thanks for the authors' feedback, clarifications, and additional supplementary experiments to further demonstrate the contributions. However, some previously stated concerns have not been adequately addressed.
> >
> > I do not doubt the in-domain performance improvements compared to the baseline RM methods. The key issues lie in efficiency and generalizability. Given that the critique capabilities of the current Cloud model entirely come from distilling stronger models, the scaling and extrapolation of this strategy are questionable.
> >
> > The authors should consider adding more task-specific discussions and customized designs for the critique generation. What changes occur when we can provide appropriate chain-of-thought processes before scoring? How do we evaluate the adaptability of this chain-of-thought component with respect to model scoring or uncertainty?
> >
> > Overall, the novelty and discussion of the current work are insufficient to improve the existing RLHF framework. To enhance the paper's contribution, more theoretical analysis or application experiments could be added.

---

### Official Review · Reviewer_T6ho · 2024-11-04

**Soundness:** 2
**Presentation:** 4
**Contribution:** 2
**Rating:** 3
**Confidence:** 4

**Summary:**

This paper proposes a reward modeling approach that combines next-token-prediction loss with binary rating loss, which improves the performance of the Llama3-8B and Llama3-70B language models on RewardBench and BoN. The authors conducted extensive ablation studies to analyze their proposed method.

**Strengths:**

- Well written, the technical approach is very clear, and Figure 1 and Figure 2 provide a clear understanding of the CLoud reward model training process.
- Based on the experimental results published by the authors, the CLoud method seems to effectively improve the model's performance on RewardBench.
- The author's ablation studies are very comprehensive.

**Weaknesses:**

- One of the most critical applications of the Reward Model is RLHF, yet the authors have not verified the improvement of CLoud RM over traditional RMs in RLHF. I believe this would greatly limit the widespread application of this method.
- It is not very clear what significant issues in Reward Modeling CLoud has addressed, such as reward overoptimization. From the experimental results, it appears to only enhance the reward model's ability to distinguish between binary responses.
- The experimental setup seems inconsistent. Based on my understanding from page 5, the authors' preference dataset and BoN dataset are derived from sampling Llama3-8B-Instruct. However, is it appropriate to use data generated by an 8B model for experiments with a 70B model? What is the rationale behind this choice? The authors should present this in the paper.
- The above point leads to my skepticism about the author's claim that "8B CLoud reward model even outperforms the 70B classic reward model."
- The experiment on self-consistency is interesting, but the author's research is not thorough enough. If they could investigate how the quality of critique generation affects scoring accuracy, and combine this with methods like COT, TOT, etc., for further exploration, it would be more impressive.

**Minors**

- I'm not particularly fond of the title "Results" for Section 3; terms like "Experiment" might be more appropriate.
- RQ2 in Section 3 seems a bit odd because there hasn't been any prior mention of on-policy/off-policy concepts, which makes it difficult to understand.

**Questions:**

- How much more overhead does CLoud RM incur during inference compared to traditional RM? Given that CLoud RM needs to generate an entire sequence before scoring, it is foreseeable that it would be more time-consuming than traditional RM. My concern about the inference cost grew even more after discovering that the author used the self-consistency method for multiple sampling averages.

---

> ### Author Response · Authors · 2024-11-19
> **Response to Reviewer T6ho (part 1/2)**
>
> We would like to thank reviewer T6ho for taking the time to engage with and review our paper, and we appreciate that the reviewer finds the paper “well-written” and containing “comprehensive ablations”. Please see below for our responses to the reviewer’s feedback.
>
>
> 1. > "One of the most critical applications of the Reward Model is RLHF, yet the authors have not verified the improvement of CLoud RM over traditional RMs in RLHF. I believe this would greatly limit the widespread application of this method."
>
>    Learning a policy (i.e., PPO) is not required for RLHF. RLHF as a framework is the process of defining a policy via a reward signal derived from human feedback. Best-of-N is one such method for defining a policy from a reward and it is actually the KL optimal policy given a reward signal [1][2]. As we demonstrate that CLoud defines a strictly better Best-of-N policy on the ArenaHard chat benchmark as compared to a classic reward model (Figure 4), our work explicitly demonstrates that CLoud is better for RLHF and thus is useful to practitioners.
>
> 2. > "It is not very clear what significant issues in Reward Modeling CLoud has addressed, such as reward overoptimization. From the experimental results, it appears to only enhance the reward model's ability to distinguish between binary responses"
>
>    Improving the ability of reward models to assign preference scores to responses is the most fundamental improvement one can make to a reward model. Reward models are explicitly trained to model human preferences so surely improving the reward model’s capability to distinguish preferred responses is solving a significant issue. CLoud also solves the problem of reward models not being able to leverage additional inference compute, which we demonstrate improves reasoning performance in Figure 7. This provides an entirely new axis of scaling for improving reward quality. Additionally, we do not only look at reward model accuracies, but we also show that CLoud leads to improved downstream policies optimized using this reward (see first response and Figure 4).
>
> 3. > "Based on my understanding from page 5, the authors' preference dataset and BoN dataset are derived from sampling Llama3-8B-Instruct. However, is it appropriate to use data generated by an 8B model for experiments with a 70B model? … The above point leads to my skepticism about the author's claim that "8B CLoud reward model even outperforms the 70B classic reward model."
>
>    We believe this is an appropriate setup as the training setup is identical between the CLoud and classic model, so the tests still isolate for preference modeling capability. Additionally, common RM training benchmarks such as the original UltraFeedback dataset [3] contain responses from smaller 8B models.
>
>    To demonstrate that our results are not just an artifact of using 8B responses, we have trained both CLoud 8B and 70B and classic 8B and 70B on the original responses in the Ultra datasets. For this experiment, we used critiques generated by GPT-4o, as we had created this dataset before the rebuttal and would not have time to generate critiques with Llama 405B. We find that CLoud models still significantly outperform classic models and that CLoud 8B recovers a large amount of the gap to the classic 70B model.
>
>     | Model | Chat | Chat Hard | Safety | Reasoning | Average |
>     | :-: | :-: | :-: | :-: | :-: | :-: |
>     | Classic 8B  | 95.25 | 54.75 | 71.76  | 83.83 | 76.40 |
>     | CLoud 8B    | 95.44 | 58.26 | 77.52  | 87.63 | 79.71 |
>     | Classic 70B | 95.81 | 69.30 | 82.70  | 88.47 | 84.07 |
>     | CLoud 70B   | 96.09 | 73.03 | 88.11  | 94.55| 87.94 |
>
> 4. > "The experiment on self-consistency is interesting, but the author's research is not thorough enough. If they could investigate how the quality of critique generation affects scoring accuracy, and combine this with methods like COT, TOT, etc., for further exploration, it would be more impressive."
>
>    While we do already investigate when self-consistency is useful by looking at its relationship with the reasoning horizon of the problem (Figure 8), we agree that investigating the impact that critique quality has on self-consistency would be interesting. However, we believe it’s out of scope given the large ground that our paper already covers. While all experiments are already conducted in the setting of CoT critiques, we agree that methods such as ToT would be interesting to investigate as exciting future directions.
>
> 5. > "I'm not particularly fond of the title "Results" for Section 3; terms like "Experiment" might be more appropriate."
>
>    We will update the section title to be “Experiments” in the final version.
>
> 6. > "RQ2 in Section 3 seems a bit odd because there hasn't been any prior mention of on-policy/off-policy concepts, which makes it difficult to understand."
>
>    Thank you for pointing this out, we will explicitly state the distinction between on and off-policy in our methods section of the final paper.

---

> > ### Author Response · Authors · 2024-11-19
> > **Response to Reviewer T6ho (part 2/2)**
> >
> > 7. > "How much more overhead does CLoud RM incur during inference compared to traditional RM? Given that CLoud RM needs to generate an entire sequence before scoring, it is foreseeable that it would be more time-consuming than traditional RM. My concern about the inference cost grew even more after discovering that the author used the self-consistency method for multiple sampling averages."
> >
> >    CLoud does introduce overhead at inference as compared to classic reward models, however, the improvement in reward modeling performance justifies this overhead. Namely, suppose one was trying to maximize the performance of their reward model and they were already using the largest base model available. To improve reward modeling performance one would need to scale some part of the system, and as model size is no longer a valid lever, one would need to scale inference compute via critiques as done in CLoud. It’s more productive to think of the overhead of CLoud as a tradeoff. Similar to the case where you are using the largest existing base model, suppose you are using the largest reward model that fits within your machine’s memory constraints. If you want to produce a better reward model subject to such constraints one way is by adding critiques as we propose.
> >
> >    We would also like to clarify that most of our results are performed without self-consistency. Performing added inference compute via self-consistency is just another axis of scaling we make available to practitioners for further reward modeling improvements.
> >
> >
> > ## References
> > [1] Gui, Lin, Cristina Gârbacea, and Victor Veitch. "BoNBoN Alignment for Large Language Models and the Sweetness of Best-of-n Sampling." arXiv preprint arXiv:2406.00832 (2024).
> >
> > [2] Beirami, Ahmad, et al. "Theoretical guarantees on the best-of-n alignment policy." arXiv preprint arXiv:2401.01879 (2024).
> >
> > [3] Cui, Ganqu, et al. "Ultrafeedback: Boosting language models with high-quality feedback." arXiv preprint arXiv:2310.01377 (2023).

---

> ### Comment · Reviewer_T6ho · 2024-11-25
>
> Thank you very much for the detailed response provided by the author. In response, I will also provide the following detailed comment:
>
> > Learning a policy (i.e., PPO) is not required for RLHF. RLHF as a framework is the process of defining a policy via a reward signal derived from human feedback.
>
> Which published papers did the author derive this statement from? If RLHF does not require learning a policy, then is the "RL" in RLHF not reinforcement learning, but rather reward learning?
>
> > Improving the ability of reward models to assign preference scores to responses is the most fundamental improvement one can make to a reward model.
>
> Improving the discernment of responses is the ultimate goal of the BT reward model, but not of the task of reward modeling itself. Reward models can take many forms, such as more dense token-level supervision. The author mentions that Cloud utilizes the language capabilities of the language model itself to improve scoring accuracy, but this process still requires a more powerful model, such as GPT-4o. When accessing GPT-4o, there are many methods to improve the scoring accuracy of the BT RM. I do not understand the necessity of Cloud if it is merely an enhanced version of the BT RM. (The enhancement here is still questionable because it does not necessarily improve the performance of methods like PPO.)
>
> > Cloud does introduce overhead at inference as compared to classic reward models, however, the improvement in reward modeling performance justifies this overhead.
>
> Instead of writing lengthy explanations, I would prefer to see the author use tabular data to present the differences in inference time consumption between Cloud RM and baseline methods.

---

> > ### Author Response · Authors · 2024-11-29
> >
> > We appreciate the reviewer's thorough feedback and continued engagement with our work. We hope that we can adequately address the reviewer's concerns with the following responses.
> >
> > > “Which published papers did the author derive this statement from? If RLHF does not require learning a policy, then is the "RL" in RLHF not reinforcement learning, but rather reward learning?”
> >
> > While we believe that BoN under a reward model constitutes RL as BoN optimizes against a reward function at inference time and RL is the process of optimizing against a reward, we don’t think it's particularly productive to debate over what is or isn’t exactly RLHF as the reviewer and authors may have slightly different opinions. However, what matters for practitioners and the field is defining policies that are better aligned with a given set of preferences, regardless of whether the policy is achieved through methods such as RLHF, DPO, or sampling such as BoN. Our results explicitly demonstrate that CLoud leads to more aligned policies when used with BoN, which is what we believe matters for the community.
> >
> > In addition to the previous works we shared that demonstrate that BoN is the KL optimal policy, we would also like to point the reviewer to Section 3.5 of Gao et al. [1] which empirically investigates the similarities between PPO and BoN. They find that when looking at proxy rm score (the score optimized against) vs. gold rm score (underlying gold reward signal), BoN and PPO optimize the reward in similar ways, suggesting that BoN optimization can serve as a good proxy for impact on PPO.
> >
> > Finally, we are currently running an experiment to test the impact of CLoud RMs on DPO. We hope that regardless of the above discussion on aligned policies, this experiment will convince the reviewer that CLoud is useful for learning aligned policies.
> >
> >
> > > “Improving the discernment of responses is the ultimate goal of the BT reward model, but not of the task of reward modeling itself. Reward models can take many forms, such as more dense token-level supervision.”
> >
> > We agree that response preference classification isn’t the ultimate goal of reward models and our first response, as the reviewer quoted, explicitly states that the purpose is to improve the quality of “assign[ing] preference scores to responses”. In our work, we performed two separate evaluations of how CLoud improves the assignment of preference scores. While one evaluation method is the performance on preference classification tasks, the other half of our evaluations investigate how the rewards from CLoud models lead to better aligned policies. Through both of these evaluations, we think that our paper robustly demonstrates that CLoud improves the quality of the rewards generated.
> >
> > > "When accessing GPT-4o, there are many methods to improve the scoring accuracy of the BT RM."
> >
> > We would like to clarify that we only use GPT-4o to generate oracle critiques as an approximation for human critiques. While we state this on the top of page 4, we will further clarify this point in our final paper. When CLoud is applied to human critiques, it is obviously no longer feasible to produce human critiques at inference.
> >
> > > “I do not understand the necessity of Cloud if it is merely an enhanced version of the BT RM.”
> >
> > We strongly disagree with the sentiment that work on improving BT RMs is unnecessary. BT RMs are commonly used in practice including in the most performant models as evidenced by their use in post-training the Gemma [2] and Llama 3 [3] series of models.
> >
> > Additionally, while we only investigate BT RMs in this work, our paper opens the door for future work on leveraging language for other types of RMs. The main innovation of CLoud, adding language to the reward modeling process, is fully compatible with Plackett-Luce, process supervision reward models, etc and we believe that it is a general method for improving reward model performance.

---

> > > ### Author Response · Authors · 2024-11-29
> > >
> > > > “Instead of writing lengthy explanations, I would prefer to see the author use tabular data to present the differences in inference time consumption between Cloud RM and baseline methods.”
> > >
> > > The point of the longer explanation was to provide the nuance for why CLoud is desirable even though it increases the overhead. CLoud increases the overhead, but we wanted to justify why increasing the overhead alone is not, in our opinion, a valid reason to reject our work. Many techniques for improving LLMs such as scaling pretraining, Chain-of-Thought, and a large number of other inference techniques also increase the overhead, but are necessary to improve performance. Given the above context, we have evaluated the run-time of CLoud and classic models on RewardBench. We will add a discussion of the overhead to the final version of the paper.
> > >
> > > The total time to score all RewardBench examples using 1x 80GB A100 is 2 minutes 5 seconds for Classic 8B and 30 minutes 9 seconds for CLoud 8B. The total time to score all RewardBench examples using 8x 80GB A100 is 3 minutes 1 second for Classic 70B and 39 minutes 8 seconds for CLoud 70B.
> > > \
> > > \
> > > Again, we would like to thank the reviewer for their continued engagement . If the reviewer has any more questions or concerns please let us know so we may address them and improve the paper.
> > >
> > >
> > > ## References
> > >
> > > [1] Gao, Leo, John Schulman, and Jacob Hilton. "Scaling laws for reward model overoptimization." International Conference on Machine Learning. PMLR, 2023.
> > >
> > > [2] Team, Gemma, et al. "Gemma: Open models based on gemini research and technology." arXiv preprint arXiv:2403.08295 (2024).
> > >
> > > [3] Dubey, Abhimanyu, et al. "The llama 3 herd of models." arXiv preprint arXiv:2407.21783 (2024).

---

### Meta-Review · Area_Chair_Yzus · 2024-12-18

**Metareview:**

The paper introduces Critique-out-Loud reward models, which generate natural language critiques before predicting scalar rewards. This approach aims to improve the reasoning and preference classification capabilities of reward models. The authors present empirical results demonstrating performance improvements on RewardBench and ArenaHard benchmarks through Best-of-N scoring using CLoud reward model.

Several reviewers found the contribution incremental, primarily combining critique generation and reward scoring without sufficient novelty. Concerns were raised regarding the lack of RLHF experiments to substantiate claims about downstream improvements. The reliance on critiques from larger models (e.g., Llama-3.1-405B) raises questions of whether the gains stem from methodology or model distillation. The authors addressed some concerns from reviewers but the doubts about the novelty and effectiveness of CLoud in RL settings beyond best-of-N remain.

**Additional Comments On Reviewer Discussion:**

During the rebuttal, reviewers raised concerns about the novelty of combining critique generation with reward scoring, the lack of RLHF experiments to validate downstream utility, and the scalability and generalizability of the method given its reliance on critiques from larger models. The authors provided detailed responses, added experiments of using a system prompt, and clarified use of larger models for critique generation. However, these responses did not sufficiently address the concerns about novelty and effectiveness of the proposed approach in RL settings beyond best-of-N.

---

### Decision · Program_Chairs · 2025-01-22

Reject